# MLE-Smith: Scaling MLE Tasks with Automated Multi-Agent Pipeline

**Rushi Qiang**[1]**, Yuchen Zhuang**[1]**, Anikait Singh**[2]
**Percy Liang**[2]**, Chao Zhang**[1]**, Sherry Yang**[2]**, Bo Dai**[1]

[1]Georgia Institute of Technology
[2]Stanford University

## Abstract

While Language Models (LMs) have made significant progress in automating machine learning engineering (MLE), the acquisition of high-quality MLE training data is significantly constrained. Current MLE benchmarks suffer from low scalability and limited applicability because they rely on static, manually curated tasks, demanding extensive time and manual effort to produce. We introduce `MLE-Smith`, a fully automated multi-agent pipeline, to transform raw datasets into competition-style MLE challenges through an efficient *generate–verify–execute* paradigm for scaling MLE tasks with verifiable quality, real-world usability, and rich diversity. The proposed multi-agent pipeline in `MLE-Smith` drives structured task design and standardized refactoring, coupled with a hybrid verification mechanism that enforces strict structural rules and high-level semantic soundness. It further validates empirical solvability and real-world fidelity through interactive execution. We apply `MLE-Smith` to 224 of real-world datasets and generate 606 tasks spanning multiple categories, objectives, and modalities, demonstrating that `MLE-Smith` can work effectively across a wide range of real-world datasets. Evaluation on the generated tasks shows that the performance of eight mainstream and cutting-edge LLMs on `MLE-Smith` tasks is strongly correlated with their performance on carefully human-designed tasks, highlighting the effectiveness of the `MLE-Smith` to scaling up MLE tasks, while maintaining task quality.

## 1 Introduction

Large Language Model (LLM) agents have demonstrated remarkable capabilities in automating complex coding and engineering domains (Chan et al., 2024; Qiang et al., 2025; Nathani et al., 2025; Jing et al., 2024; Yang et al., 2024; Jimenez et al., 2023), with machine learning engineering (MLE) emerging as a key frontier for evaluating the capability of models today. The development of sophisticated MLE agents, capable of autonomously handling tasks from data pre-processing to model tuning and deployment, promises to revolutionize scientific discovery and industrial applications. However, evaluating and developing such agents poses a significant challenge, due to the inherent complexity of MLE workflows, the need for domain-specific knowledge, and the iterative, feedback-driven nature of real-world machine learning pipelines. Developing robust MLE agents, therefore, requires not only the design and implementation of agent frameworks but also the creation of holistic environments and benchmarks that support end-to-end experimentation and structured evaluation under truly real-world conditions, encompassing diverse task distributions.

Recent efforts have established valuable benchmarks and interactive environments for evaluating and training these agents (Huang et al., 2023; Jing et al., 2024; Chan et al., 2024; Qiang et al., 2025; Nathani et al., 2025). However, existing benchmarks such as MLE-Bench (Chan et al., 2024) and DS-Bench (Jing et al., 2024) and gym-like interactive environments such as MLE-Dojo (Qiang et al., 2025) and MLGym (Nathani et al., 2025) offer only static collections of tasks, and their construction remains heavily reliant on extensive human curation. This manual effort stems from two main sources: (1) the competitions selected for inclusion in these benchmarks are often carefully designed by human experts, and (2) the benchmarks require substantial engineering work to adapt

these competitions into a standardized format suitable for benchmarking. Such adaptation typically involves non-trivial engineering efforts such as the pre-processing and splitting of data into train and test splits, along with implementing evaluation scripts and establishing a scoring mechanism. In addition, the ambition to establish a comprehensive environment for evaluating and training MLE agents imposes further demands on the scale and diversity of available MLE tasks. The continued reliance on static, manually curated tasks restricts the diversity and realism of interaction scenarios and introduces a scalability bottleneck that impedes the rapid development and reliable assessment of next-generation MLE agents. Thus, overcoming this limitation necessitates an automated framework that can continuously generate, verify, and evolve MLE tasks at scale.

Building such a framework for scaling MLE tasks presents a formidable challenge: how can the framework rigorously validate the correctness and practical value of each newly generated task? Unlike conventional supervised datasets, an MLE benchmark must satisfy multiple intertwined criteria: (i) *Structural integrity*, ensuring that all associated components including data pre-processing scripts, file directory hierarchies, and evaluation pipelines, must execute end-to-end without manual intervention, ensuring that the task is reproducible and computationally viable; (ii) *Semantic soundness*, confirming that the defined learning objective must be coherent, and the input–output structure must reflect the natural affordances and signals present in the source dataset, avoiding degenerate or trivial mappings; and (iii) *Empirical solvability*, demonstrating that the task should be non-trivial yet tractable—i.e., standard baseline agents must be able to achieve meaningful performance and exhibit stable improvement under reasonable training protocols. A failure on any of these dimensions undermines the utility of the task, preventing it from eliciting meaningful behavioral differences across agents or supporting their effective training and development in interactive settings.

To address these challenges, we present `MLE-Smith`, a fully automated framework that transforms raw datasets into competition-style MLE tasks through a scalable *generate–verify–execute* pipeline. `MLE-Smith` is carefully designed to enforce structural integrity, semantic soundness, and empirical solvability by integrating a **multi-agent generation workflow**, a robust hybrid verification mechanism, and an execution-based validation loop, as illustrated in Figure 1, which provides an overview of the end-to-end paradigm. The system features three specialized agents—Brainstormer, Designer, and Refactor—that generate, concretize, and standardize task proposals in a modular, auditable manner. A persistent verification mechanism, combining both deterministic checks and agent-based reviews, continuously ensures the correctness and coherence of tasks. Finally, each task is validated by interactive execution between a validation MLE agent and MLE environments, confirming that it supports end-to-end execution and delivers non-trivial signals on the performance of ML solutions. This principled pipeline ensures that each generated task is format-consistent, executable, and verifiable, while remaining practically meaningful for training and evaluating MLE agents.

We summarize our main contributions as follows:

- **A fully automated task generation framework.** We propose `MLE-Smith`, the first end-to-end system that transforms raw datasets into competition-style machine learning engineering (MLE) tasks through a scalable *generate–verify–execute* pipeline. Unlike prior efforts that rely on static curation, `MLE-Smith` enables continuous generation of realistic and diverse MLE challenges at scale, without *any* human intervention.

- **A hybrid verification mechanism.** To ensure the quality and utility of generated tasks, we design a multi-layer verification mechanism that combines static format validation, semantic alignment, and execution-based tests of empirical solvability. This hybrid stack enforces rigorous guarantees on task integrity, ensuring that each constructed challenge is well-structured, executable, and grounded in realistic machine learning scenarios.

- **A large-scale, diverse generated task suite.** We apply `MLE-Smith` to 224 real-world datasets and produce 606 fully verified tasks spanning a wide spectrum of modalities (e.g., tabular, vision, time series), learning objectives (e.g., classification, regression, ranking), and domains (e.g., healthcare, sports). Evaluation on a representative subset of 50 tasks with eight cutting-edge LLMs reveals strong correlation with rankings of these LLMs on human-curated benchmarks, demonstrating that `MLE-Smith` yields challenging, discriminative, and generalizable tasks suitable for evaluating and eventually training next-generation MLE agents.

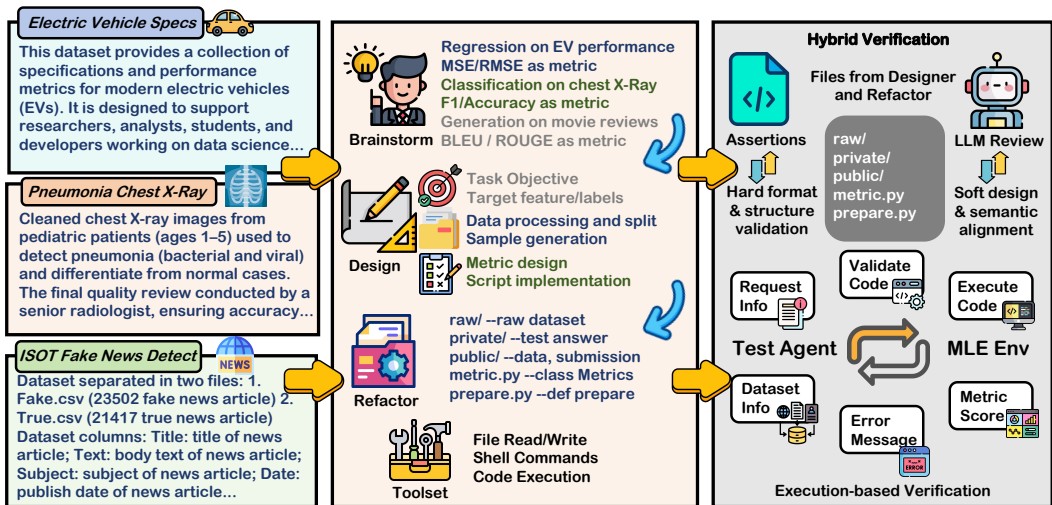

Figure 1: `MLE-Smith` automatically generates competition-style machine learning engineering (MLE) tasks from raw datasets through a *generate–verify–execute* paradigm.

## 2 RELATED WORKS

**Agent Benchmarks and Environments.** Recent efforts have introduced a diverse suite of benchmarks and interactive environments for the evaluation and development of LLM-based agents across multiple domains, including software engineering (SWE) benchmarks (Jimenez et al., 2023; Pan et al., 2024a; Yang et al., 2024; Zhang et al., 2025; Zan et al., 2025; Aleithan et al., 2024) that test agents' ability to modify large codebases and repair real-world bugs, web navigation and browsing tasks (Chezelles et al., 2024; Zhou et al., 2023; Pan et al., 2024b; Levy et al., 2024; Wei et al., 2025; Wu et al., 2025; Yao et al., 2022) that evaluate agents' capacity to navigate complex websites or device interfaces, deep research settings (Du et al., 2025; Bosse et al., 2025; Phan et al., 2025) that require multi-step reasoning and information aggregation, general tool-use environments (Yao et al., 2024; Qin et al., 2023; Mialon et al., 2023; Liu et al., 2023; Luo et al., 2025) that probe agents' ability to orchestrate diverse tools and external resources, and studies of human–agent collaboration in dynamic task scenarios (Shao et al., 2024). In the MLE domain, a growing body of testbeds assesses agents on end-to-end workflows. For example, MLAGENTBENCH (Huang et al., 2023) offers 13 curated MLE tasks with baselines and performance thresholds, MLE-BENCH (Chan et al., 2024) standardizes 75 Kaggle competitions for structured MLE evaluation, DS BENCH (Jing et al., 2024) includes 74 modeling tasks reflecting realistic data science processes, MLGYM (Nathani et al., 2025) provides a Gym-style suite for AI research workflows, and MLE-DOJO (Qiang et al., 2025) scales to over 200 fully executable MLE tasks with step-wise interaction. While these MLE platforms advance realism and breadth, they remain limited by finite, manually curated task sets. In contrast, `MLE-Smith` proposes a fully automated framework for scalable and high-quality MLE task generation, which allows for the continual generation of novel tasks in the MLE domain.

**Automated Task Generation.** Automated task generation has emerged as a promising direction for scaling agent evaluation and training. TASKCRAFT (Shi et al., 2025) creates scalable, multi-tool agentic tasks with execution traces via compositional extensions. AUTOCODEBENCH (Chou et al., 2025) generates high-difficulty, multilingual code problems with LLM-driven reverse synthesis and test validation. SWE-SMITH (Yang et al., 2025) synthesizes tens of thousands of bug-inducing software engineering tasks from real-world Python repositories. SELF-CHALLENGING (Zhou et al., 2025) trains agents to generate and solve their own Code-as-Task problems with built-in verification, enabling high-quality self-supervised RL. SQLM (Chen et al., 2025) frames task generation as asymmetric self-play, where models propose and solve increasingly challenging problems without external data. `MLE-Smith` serves as the first automated framework for task generation in the MLE domain, paving the way for scalable agent evaluation and training on realistic, high-quality tasks.

# 3 METHODS

`MLE-Smith` automatically generates competition-style machine learning engineering (MLE) tasks from raw datasets (from sources such as Kaggle) through a *generate–verify–execute* paradigm. The pipeline couples (i) **structured multi-agent generation** that designs and generates feasible tasks in multiple directions, (ii) a **hybrid verification mechanism** that enforces both hard structural constraints and soft semantic criteria, and (iii) **execution-based validation** inside an interactive MLE environment to ensure empirical solvability and real-world validity. This sequential architecture is designed to balance the diversity of task proposals with strong guarantees on the structural correctness and downstream usability of generated MLE tasks.

## 3.1 MULTI-AGENT GENERATION WORKFLOW

`MLE-Smith` employs three specialized agents that handoff generated artifacts in a sequential pipeline augmented with controlled feedback loops to allow for upstream refinement. Each agent has access to useful domain tools, including file I/O, shell commands, code execution, and always generates outputs in a pre-defined, structured format amenable to automated verification. The middle part of Figure 1 illustrates how these agents sequentially advance the pipeline and produce the corresponding deliverables.

**Brainstormer.** Given a dataset overview along with the toolset for in-depth, multi-round data exploration, the Brainstormer enumerates a set of candidate task formulations rather than a single design, recognizing that a single dataset often supports multiple plausible learning objectives and modeling strategies. This diversity-aware generation allows the system to fully exploit the dataset's potential. The number of candidate tasks is adaptively determined by the Brainstormer based on the dataset's intrinsic properties and structural characteristics. A key principle is that all labels and features must be accurate and grounded in the data itself, either explicitly provided or deterministically derived, rather than synthetic or heuristically constructed. Each proposal specifies candidate **prediction targets** (classification labels, regression variables, sequence outputs), **evaluation metrics** (e.g., accuracy, macro-F1, RMSE, or domain-specific scores), **data utilization** (e.g., preprocessing, feature construction, label extraction) and **justifications** that articulate the rationale and practical usability of the proposed design. Equipped with domain tools, the Brainstormer gains comprehensive and in-depth insights, enabling it to generate grounded and valuable task proposals. By explicitly separating hypothesis generation from commitment, `MLE-Smith` preserves design optionality and encourages diversity without sacrificing feasibility.

**Designer.** For each candidate task formulation, the Designer is responsible for instantiating a fully specified machine learning engineering (MLE) task that can be executed end-to-end without manual intervention. This includes constructing 4 components necessary to define, prepare, and evaluate the task in a reproducible and verifiable manner: (i) preprocessing the raw dataset and producing deterministic training and test splits with appropriate label coverage and data integrity guarantees; (ii) defining input and output schemas that govern the structure of model predictions and evaluation targets; (iii) specifying the evaluation protocol and instantiating a fair, task-specific metric that captures performance with numerical stability; and (iv) generating the complete suite of auxiliary components, including task descriptions that summarize the problem setup, data usage, and evaluation strategy; preparation scripts that performs data preprocessing, splitting, and validation checks; structured sample submission files with randomized and valid predictions; evaluation scripts for submission format validation and metric score calculation; and testing scripts to verify the correctness and consistency of the generated scripts.

Together with the original dataset, these artifacts form a complete, self-contained MLE task package that can be executed, evaluated, and iterated upon by agents in an interactive environment. Generating multiple such packages in parallel allows for efficient exploration of diverse task designs and principled comparisons across candidate formulations.

**Refactor.** The Refactor module standardizes all candidate task designs into a unified and well-specified format. We present the details of this structural task format in Appendix A.2. Rather than merely cleaning code or reorganizing files, this stage rewrites each task into a shared, consistent schema that defines the preparation interface, input/output specifications, metric implementation, canonical file structure, and feedback reporting mechanism. We define a set of conventions that

govern the structure and semantics of valid tasks paired with verification routines that check conformance to these standards. By enforcing these common conventions while preserving task-specific logic, the Refactor ensures format consistency, cross-file coherence, and reliable execution. This unified representation enables downstream validation of structural correctness, streamlining automated testing pipelines to verify whether each task executes end-to-end without intervention.

## 3.2 HYBRID VERIFICATION MECHANISM

To guarantee that every generated task is not only correct in terms of format but also semantically coherent and practically solvable, we implement a persistent *Hybrid Verification Mechanism*—a multi-layered, multi-agent collaborative contract through the entire *generate–verify–execute* pipeline. This mechanism executes across stages and comprises three complementary verification strategies: deterministic *Assertions*, model-mediated *Reviews*, and empirical *Execution-based Validation*.

**Assertions (deterministic guards).** Assertions encode mandatory structural constraints that are enforced through deterministic checks. These include validation of existing files, directory layout, and compliance with a structured schema for functions, classes, and scripts. Crucially, each assertion stage serves as a gatekeeper, ensuring that downstream modules can operate reliably without encountering missing inputs or malformed artifacts. Prior to Refactor, Assertions confirm the completeness and structural integrity of outputs from the Designer. As a representative example, Pre-Refactor Assertions may verify that the *metric.py* and *prepare.py* scripts execute correctly, and that both a *sample submission* and a corresponding *test answer* are successfully created. Post-Refactor, Assertions enforce full conformance to the unified task schema, including function signatures, interface formats, and execution scripts. For instance, they may examine whether the entire directory satisfies the pre-defined, unified format as in Appendix A.2. These rigid checks not only eliminate syntactic and structural defects but also ensure that the task satisfies all requirements for automated downstream execution. A task that successfully passes all assertions can be regarded as a fully structured and automation-ready MLE task, capable of running end-to-end without human intervention.

**Reviews (semantic validation).** Where assertions enforce formal correctness, Reviews evaluate the semantic quality and intent alignment of each task. Leveraging an LLM-based agent as the reviewer, this stage assesses the clarity of task descriptions, the appropriateness of metrics, and whether the setup encourages meaningful agent behavior over shortcut solutions. For example, Reviews may flag task descriptions that omit necessary information, or ones that leak ground truths, which would pass assertions but compromise semantic validity. Though non-deterministic, Reviews serve as a soft but crucial layer that guides refinement when rigid rules are insufficient.

**Execution-based validation (empirical tractability).** Beyond structural and semantic checks, a well-posed MLE task must also demonstrate empirical viability: it should admit learnable patterns, enable meaningful performance differentials, and support full-pipeline execution under realistic agentic interactions. To verify this, we introduce *execution-based validation stage* that runs the entire task within an interactive MLE environment. This stage leverages a coding agent with action budgets to simulate a typical MLE agent interaction process. The environment, based on MLE-Dojo (Qiang et al., 2025), exposes an API for retrieving task metadata, validating code, executing scripts, and evaluating submissions. This interface allows for transparency over the actions that the step-wise agent takes and provides fine-grained feedback on execution results and performance.

The environment monitors two key aspects of empirical validation: (*i*) *realistic pipeline validation*, which ensures that the full pipeline, including data preparation, model training, evaluation and scoring, executes successfully without human assistance; and (*ii*) *performance validation*, which verifies that test agents achieve non-trivial predictive performance and that the evaluation metric exhibits sensitivity to method quality. Failures along either dimension are logged as structured defects and routed back into the verification mechanism, triggering either targeted refinement by the Refactor or Designer module or a re-execution of the corresponding stage. Positioned at the end of the generation pipeline, execution-based validation ensures empirical solvability by running the full task pipeline and measuring non-trivial agent performance. It captures failure modes that escape earlier static or semantic checks, serving as the ultimate safeguard for real-world usability.

Taken together, the three layers of verification offer distinct but complementary guarantees: *Assertions* ensure structural correctness, *Reviews* ensure semantic alignment, and *Execution* ensures real-world solvability and usability. Only tasks that satisfy all three criteria are retained as verified, high-quality MLE challenges suitable for automated benchmarking and agent development.

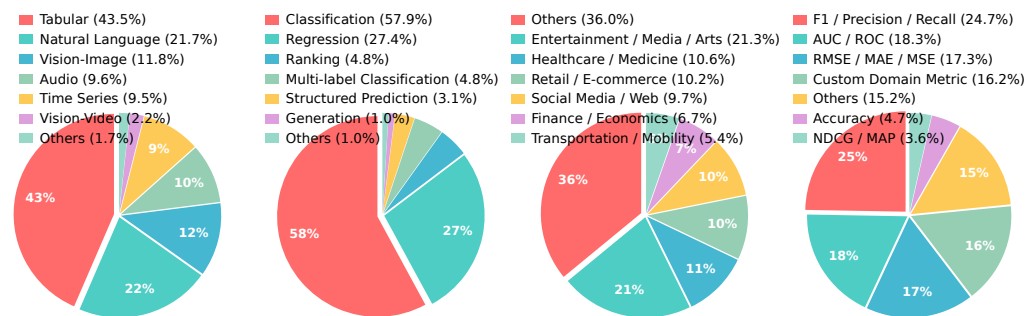

Figure 2: Domain, Modality, and Formulation Distribution of `MLE-Smith` generated tasks. From left to right, the panels show the distributions of modality, objective, domain, and metric, respectively. "Others" category aggregates all types whose individual proportions are relatively minor.

**Empirical Justification of Pipeline Components.** We conduct a comprehensive ablation study in Appendix B to empirically justify the necessity of each component in the multi-agent pipeline and verification mechanism. The results confirm that each component plays an essential and complementary role in achieving an effective balance of cost, performance, and output quality.

## 4 AUTOMATED TASK GENERATION

`MLE-Smith` can operate seamlessly across datasets of diverse modalities, formats, and domains. To comprehensively evaluate the performance and capabilities of `MLE-Smith`, we collect datasets from Kaggle, the most large-scale platform that hosts diverse, real-world machine-learning competitions and data resources. We sampled 224 datasets from those with high usability scores as the experimental corpus and generated 606 tasks from these 224 source datasets. We reserve a subset of 50 generated tasks to evaluate the quality of `MLE-Smith`, by measuring the alignment of the performance of mainstream LLMs with the MLE-Dojo leaderboard.

### 4.1 AGENT AND ENVIRONMENT SETUPS

We use GPT-5 (OpenAI, 2025a) to serve as the backbone model for all of the agents in `MLE-Smith`. We use a default temperature of 1.0 for GPT-5. We emphasize that the proposed multi-agent pipeline is compatible with any LLM. For each dataset, the Brainstormer Agent is allowed up to 30 steps of tool-call actions. Additionally, for each source dataset, the Brainstormer Agent is allowed to brainstorm at most 3 candidate task formulations. Then, for each candidate, both the Designer and Refactor Agents have at most 3 retry times to pass all assertions. For every proposed task formulation, the Designer and Refactor are additionally allocated a separate budget of up to 30 steps to complete their respective processes. For the execution-based validation stage, we adapt MLE-Dojo and set up an interactive MLE environment with `request_info` and `execute_code` interfaces, which respectively support retrieving task-related information and evaluating submissions. The environment provides step-wise, structured feedback to agents. We implement a ReAct-style MLE Agent (Yao et al., 2023; Sun et al., 2023) with a budget of up to 10 steps to generate, debug, and execute code submissions to get valid metric scores.

### 4.2 STATISTICS OF GENERATED TASKS

**Scale and Cost.** `MLE-Smith` produced a total of 606 fully verified tasks across 224 distinct source datasets, demonstrating both scalability and efficiency of our proposed approach. On average, each dataset yields 2.71 competition-style tasks with the end-to-end preparation time per task averaging 419.98 seconds, and per dataset averaging 1136.20 seconds. This runtime excludes the execution-based verification stage, as this stage depends heavily on dataset/task characteristics, hardware configuration (GPU & CPU), and the diversity of agent-generated code, exhibiting large variance. Here, the per-task execution time is typically below 600 seconds. The overall pipeline incurred an average cost of $0.78 per task and $2.11 per dataset, including all the generation workflow and verification stages. This time required for automatic task generation is substantially lower than the manual cost of human experts authoring competition-style tasks, and also significantly less than the engineering effort needed to localize and standardize Kaggle competitions into benchmark-ready formats. Moreover, the execution-based verification stage is negligible when compared to the time

it would take for human practitioners to solve a task and achieve a meaningful score. This considerable efficiency in time strongly underscores the scalability of `MLE-Smith` for large-scale machine learning engineering (MLE) task generation.

**Domain, Modality, and Formulation Diversity.** The generated tasks span a broad spectrum of real-world data modalities, target objectives, task domains and evaluation metrics. Figure 2 illustrates the detailed distributions of generated tasks in these four aspects. Specifically, the task modalities of `MLE-Smith` generated tasks includes Tabular, Image, Video, Audio, Natural Language, Time Series, and other structured sources. Due to the characteristics of the source Kaggle datasets, tabular and natural language modalities appear more frequently. However, other modalities also constitute a substantial portion of the generated tasks. The benchmark covers a variety of formulations: while classification and regression are relatively common, it also includes ranking, multi-label classification, structured prediction, and generation tasks, offering diverse challenges for MLE agents. Compared to modality and objective, metric design tends to exhibit greater flexibility, as it is not necessarily tied to the intrinsic properties of the dataset. Thus, `MLE-Smith` naturally reflects this flexibility. The benchmark employs a wide range of evaluation metrics, with F1, precision, and recall collectively accounting for 24.7%, followed by AUC/ROC (18.3%), RMSE/MAE/MSE (17.3%), and a notable portion of custom domain-specific metrics (16.2%). Other metrics, such as ranking-based measures like NDCG and MAP (3.6%), further contribute to the overall diversity, highlighting the pipeline's ability to support nuanced evaluation tailored to different task types.

**Applicability Beyond Well-Structured Datasets.** While many Kaggle datasets are relatively well-organized, they are not universally well-structured or pre-cleaned. To demonstrate the generality of `MLE-Smith`, we additionally evaluate it on three categories of notably raw datasets: unprocessed tabular data (no predefined features or labels), raw server logs, and raw scientific sensor data. In all cases, `MLE-Smith` autonomously organizes the data, defines appropriate features and labels, and produces validated tasks, confirming its applicability beyond competition-ready datasets. Full details and generated task descriptions are provided in Appendix E.

**Agent-Wise Performance.** For each candidate formulation proposed by the Brainstormer, both the Designer and Refactor components are allowed up to three retries, with a maximum step limit imposed for each attempt. For different datasets and formulations, the number of retries and steps used by the Designer and Refactor components is summarized by the following statistics. In over 99% of cases, the Designer succeeds on the first attempt and passes all assertion checks. Approximately 92% of the time, it completes the task in no more than 15 steps, with the shortest successful case requiring only 8 steps, and none exceeding 26 steps. In contrast, the Refactor component requires more retries and tends to take more steps: around 6% of tasks are only completed successfully on the second attempt, and about 1% require a third. Across all tasks and formulations, Refactor consistently uses more than 13 steps, with the majority of tasks densely utilizing 15 to 22 steps. These results align with the intended roles and design of the agents: the Refactor typically requires more actions than the Designer, as it must read the provided examples, analyze how to standardize the code and file structure to meet the required specifications, and ultimately ensure all tests pass.

## 5  EXPERIMENTS: TASK EVALUATION

We evaluate whether the tasks generated by `MLE-Smith` faithfully reflect the difficulty and discriminative structure of real, human-designed tasks. We conduct a comprehensive evaluation of eight cutting-edge large language models (LLMs) on a curated benchmark of 100 machine learning engineering (MLE) tasks, which we refer to as the **Combined set**. This evaluation suite comprises 50 tasks from the original MLE-Dojo evaluation set **Dojo set** and 50 tasks automatically generated by `MLE-Smith` **Smith set**. Both subsets are designed to span a diverse range of data modalities, application domains, and task formulations, providing a sufficiently diverse MLE testbed.

### 5.1  EXPERIMENT SETUPS

**LLMs for Evaluation.** We consider eight cutting-edge LLMs in the evaluation and improvement of LLMs as MLE Agents on **Combined set**. Specifically, we consider `gpt-4o-mini` (2024-07-18) (Hurst et al., 2024), `gpt-4o` (2024-11-20) (Hurst et al., 2024), `o3-mini` (2025-01-31) (OpenAI, 2025b) and `o4-mini` (2025-04-16) (OpenAI, 2025c) from OpenAI, `Gemini-2.5-Flash` (Comanici et al., 2025) and `Gemini-2.5-Pro` (Comanici et al.,

Table 1: Elo ratings of eight LLMs across different categories on the Dojo set, Smith set, and Combined set. For all columns, higher scores indicate better performance. The highest score in each category is highlighted in bold, and odd-numbered rows are shaded for visual clarity.

| Model | MLE-Dojo | | | | | MLE-Smith | | | | | MLE-All |
|---|---|---|---|---|---|---|---|---|---|---|---|
| | MLE-Lite | Tabular | NLP | Vision | Overall | Vision | NLP/Tab. | Audio | Video | Overall | Combined |
| Gemini-2.5-Pro | **1272.0** | **1187.8** | **1303.6** | **1320.7** | **1254.6** | **1346.9** | 1000.7 | **1318.7** | **1484.1** | **1179.7** | **1214.3** |
| Gemini-2.5-Flash | 1189.7 | 1004.3 | 1254.5 | 1194.8 | 1146.7 | 1202.5 | 1009.1 | 1142.3 | 963.5 | 1079.3 | 1111.3 |
| o4-mini | 1019.9 | 1013.8 | 1173.2 | 1194.8 | 1068.0 | 1075.6 | **1083.5** | 1168.0 | 1114.6 | 1097.6 | 1082.9 |
| DeepSeek-Reasoner | 1095.6 | 1101.0 | 915.7 | 1122.5 | 1064.8 | 1243.8 | 1028.9 | 1030.6 | 963.5 | 1059.1 | 1061.8 |
| o3-mini | 1017.3 | 1004.3 | 1004.6 | 1043.6 | 1011.9 | 1007.1 | 1017.6 | 984.7 | 936.7 | 1003.3 | 1007.6 |
| DeepSeek-Chat | 975.4 | 976.0 | 1024.7 | 1037.4 | 990.7 | 956.2 | 1066.0 | 1055.3 | 999.5 | 1030.2 | 1011.2 |
| GPT-4o | 770.9 | 877.9 | 761.4 | 555.7 | 776.5 | 618.4 | 932.3 | 681.3 | 806.5 | 808.8 | 794.1 |
| GPT-4o-mini | 659.3 | 834.9 | 562.2 | 530.5 | 686.7 | 549.5 | 861.9 | 619.0 | 731.5 | 742.0 | 716.8 |

Figure 3: Pairwise win–loss matrices of eight models on the Dojo, Smith, and Combined sets. Each cell $(i, j)$ records the number of tasks on which model $i$ outperforms model $j$, and the aggregated score is computed by awarding $1$ point for a win, $0.5$ point for a tie, and $0$ points for a loss.

2025) from Google, and DeepSeek-V3.1-Chat (2025-03-24) (DeepSeek, 2025) and DeepSeek-V3.1-Reasoner (DeepSeek, 2025) from DeepSeek as evaluation backbone LLMs. For non-reasoning models, we set temperature=0.0 and top-$p = 1.0$. For reasoning models, we use default model settings. We take the best performance of two runs per task per model.

**Agent and Environment Design.** We implement the MLE Agent following the MLE-Dojo framework, which utilizes native actions and interacts with the MLE environment. For each task and each run, the agent is allowed up to 15 action steps and a maximum of 12 hours of execution time. The context and maximum output lengths are determined by the properties of the underlying model.

**Evaluation Metrics.** Each task is associated with a specific evaluation metric, which is used to compute the raw performance score for that task. To ensure comprehensive evaluation and allow for a fair comparison across different models, we adopt *Elo* ranking (Chiang et al., 2024) as the primary comparative indicator. We follow Chatbot Arena (Chiang et al., 2024) and estimate Elo scores by fitting a Bradley–Terry-style logistic model via maximum likelihood, using sample-weighted pairwise outcomes (wins/losses with ties treated as symmetric half-wins). We adopt a base-10 log-odds parameterization scaled to the Elo convention (scale = 400, base = 10, offset = 1000).

## 5.2 MAIN RESULTS

We compute modality-level Elo ratings on three disjoint sets: **Dojo set** (50 real tasks in MLE-Dojo), **Smith set** (50 MLE-Smith generated tasks), and **Combined set** (all 100 tasks). Table 1 presents ELO scores for all eight LLMs across different categories and task sets. Across all subsets, Gemini-2.5-Pro establishes a clear performance frontier, maintaining top rankings in almost every modality and transferring its advantage seamlessly from real to generated benchmarks. A second tier emerges with DeepSeek-V3.1-Reasoner and o4-mini, which show competitive balance across modalities: o4-mini is particularly strong on language-oriented tasks, while DeepSeek-V3.1-Reasoner delivers more robust vision performance. In contrast, the GPT-4o family consistently lags behind, especially on vision input, underscoring persistent challenges in multimodal generalization. Overall, we observe a consistent ranking trend across real and synthetic

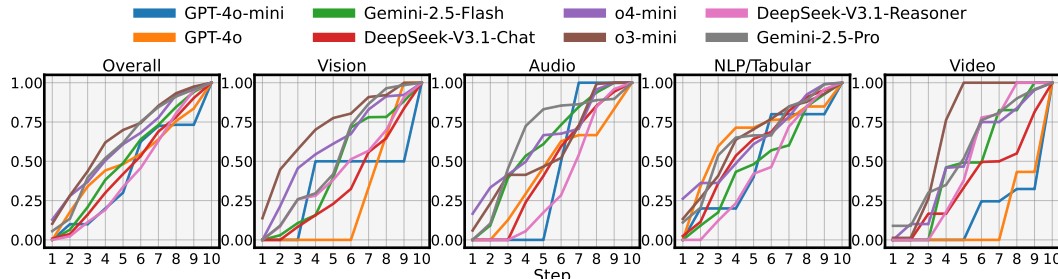

Figure 4: Step-wise Performance Dynamics of normalized raw scores. Curves are obtained by point-wise averaging over tasks in corresponding categories. Information-requesting steps are excluded.

tasks, validating the use of generated benchmarks for model differentiation. The Elo distribution also highlights the diversity of task difficulty and model specialization across input modalities.

## 5.3 STEP-WISE PERFORMANCE DYNAMICS

We study step-wise performance dynamics across different models to reveal consistent improvement patterns that reflect desirable properties of the automatically generated tasks. We exclude information-requesting steps of agents and denote the remaining steps as $u \in \{1, \ldots, 10\}$. Since realistic leaderboards and human performances are not available for generated tasks, we implement a normalization mechanism to model step-wise improvement. For each (task $t$, model $m$), raw scores are extracted from execution feedback of execute_code actions and normalized in a metric-aware manner depending on whether higher or lower values indicate better performance. Detailed formulas are provided in Appendix A.3. After normalization, missing entries are imputed, and we construct a best-so-far trajectory via a prefix maximum, yielding a nondecreasing length-10 curve per (task, model). Category-level and overall curves in Figure 4 are obtained by averaging across task trajectories. Across all categories, models exhibit consistent upward trajectories, indicating that agent performance reliably improves with steps. This trend suggests that MLE-Smith-generated tasks are learnable, provide sufficient resolution to differentiate between modeling approaches, and support iterative refinement and methodical exploration. These observations provide empirical justification for using MLE-Smith-generated tasks in the evaluation and development of MLE agents.

## 5.4 REALISM AND QUALITY OF GENERATED TASKS

To evaluate the realism and discriminative fidelity of tasks generated by MLE-Smith, we analyze the statistical alignment between model-level Elo scores computed on Dojo set, Smith set, and Combined set. Specifically, we adopt complementary statistics that capture distinct notions of agreement: (i) **linear correlation** (Pearson (Pearson, 1895)) to quantify similarity in absolute Elo magnitudes, (ii) **rank agreement** (Spearman (Spearman, 1961), Kendall (Kendall, 1938)) and **head-of-leaderboard overlap** (Top-$k$) to assess stability of model ordering, (iii) **scale and bias agreement** (Lin's Concordance Correlation Coefficient (Lawrence & Lin, 1989), *CCC*, and Bland–Altman analysis (Bland & Altman, 1986)), and (iv) **multi-rater reliability** (Cronbach's $\alpha$ (Cronbach, 1951), ICC (Shrout & Fleiss, 1979)) to test whether different Elo sets function as interchangeable evaluators over the same population. We include the details of these measurements in Appendix A.5.

Table 2: Elo agreement with complementary statistics. CCC denotes Lin's concordance correlation coefficient; Kendall $\tau_b$ accounts for ties.

| Pair | Pearson $r$ | $R^2$ | Spearman $\rho$ | Kendall $\tau_b$ | CCC | Top-3 / Top-5 |
|------|-------------|-------|-----------------|------------------|-----|---------------|
| Dojo–Smith | 0.982 | 0.964 | 0.952 | 0.857 | 0.958 | 1.0 / 0.8 |
| Dojo–Combined | 0.996 | 0.992 | 0.976 | 0.929 | 0.989 | 1.0 / 0.8 |
| Smith–Combined | 0.995 | 0.990 | 0.976 | 0.929 | 0.989 | 1.0 / 1.0 |

Across all pairs, linear relationships remain near-perfect: Dojo–Smith $r = 0.982$, Dojo–Combined $r = 0.996$, and Smith–Combined $r = 0.995$ ($R^2 = \{0.964, 0.992, 0.990\}$). Rank order is likewise stable with Spearman $\rho = \{0.952, 0.976, 0.976\}$ and Kendall $\tau_b = \{0.857, 0.929, 0.929\}$; top rankings nearly coincide (Top-3 overlap $= 1.0$ for all, Top-5 $= \{0.8, 0.8, 1.0\}$). Beyond correlation, numerical agreement is strong: CCC $\{0.958, 0.989, 0.989\}$, negligible Bland–Altman bias, and limits of agreement of roughly $\pm 96$, $\pm 51$, and $\pm 45$ Elo. Treating the three sets as interchangeable evaluators yields $\alpha = 0.993$ and ICC$(2, 1) = 0.981$, indicating excellent inter-set reliability. These

statistics consistently indicate that the Elo distributions induced by `MLE-Smith` are statistically indistinguishable from those of human–designed benchmarks, demonstrating that `MLE-Smith` effectively generates tasks with realistic difficulty and practical usability, faithfully mirroring the discriminative structure of real MLE competitions and supporting MLE agent development at scale.

**Human Evaluation.** To complement the statistical analysis, we conduct a human evaluation with ML experts from multiple institutes. We sample 50 tasks and apply a three-step inspection procedure: (1) re-running the full preparation and evaluation scripts to confirm executability; (2) checking for logical flaws, such as unsolvable setups (e.g., test labels absent from training data) or tasks admitting trivial high-score solutions (e.g., label leakage); and (3) implementing or reviewing a solution to verify that it produces a non-trivial and reasonable metric score. The evaluation found the overall task quality to be satisfactory, with no major errors identified across the 50 inspected tasks.

**Diversity of Problem-Solving Strategies.** We further analyze the solution strategies employed by agents when solving `MLE-Smith` tasks. As detailed in Appendix C, the models used by agents span multiple modalities and a wide range of complexity: approximately 40% are custom architectures (CNN, RNN, Seq2Seq, UNet, etc.), 25% are traditional ML models (logistic regression, gradient boosting, SVM, etc.), 17% are vision CNN backbones (ResNet, EfficientNet, ConvNeXt), and the remainder includes detection, segmentation, video, NLP transformer, and audio/speech models. The data processing techniques are similarly diverse, covering operations from TF-IDF vectorization and image augmentation to audio resampling and mel spectrogram generation. This diversity, which in some aspects exceeds that of MLE-Dojo (particularly in video and audio modality coverage), confirms that the generated tasks require genuinely varied problem-solving strategies rather than a homogeneous set of approaches.

## 5.5 TASK DIFFICULTY AND ERROR ANALYSIS

To further characterize the challenge posed by `MLE-Smith` tasks, we analyze agent error rates across modalities and the distribution of error types. Video and audio tasks exhibit the highest overall error rates (44.38% and 43.40%), followed by vision (39.75%) and NLP/tabular (33.06%). These nontrivial rates arise because (1) the generated tasks are novel formulations that agents have not encountered before, forcing trial-and-error exploration, and (2) each task requires correct implementation across the full MLE pipeline, making it easy to fail at any stage. Video and audio modalities, which were underrepresented in prior benchmarks, expose particularly notable weaknesses in current agents. As shown in Table 3, the error type distribution is broad, spanning value errors, data processing, imports, file handling, and model-specific issues, indicating that the tasks exercise diverse engineering competencies rather than a single failure mode.

Table 3: Distribution of error types encountered by agents on `MLE-Smith` tasks.

| **Type** | Val/Typ/Key | Import/API | DataProc | File/Path | Model/ML | Submission | Attribute | Syntax | Other |
|---|---|---|---|---|---|---|---|---|---|
| **%** | 22.29 | 18.75 | 13.62 | 11.03 | 9.79 | 7.78 | 6.31 | 5.42 | 5.00 |

## 6 CONCLUSION

We introduce `MLE-Smith`, a fully automated multi-agent pipeline for transforming raw datasets into competition-style machine learning engineering tasks. Through a principled *generate–verify–execute* paradigm, `MLE-Smith` scales task generation while ensuring structural integrity, semantic soundness, and empirical solvability. Applied to hundreds of real-world datasets, it produces a large and diverse suite of high-quality tasks that strongly correlate with human-designed benchmarks, demonstrating that generated tasks can match real competitions in realism and discriminative power.

## ACKNOWLEDGMENTS

We thank the anonymous reviewers and area chairs for their valuable feedback. This work was supported in part by ONR (N000142512173), NSF ECCS (2401391), and NSF IIS (2403240), and

also supported in part by computing resources received from the National Supercomputing Center (CSCS) and the Swiss AI initiative.

## ETHICS STATEMENT

This work adheres to the ICLR Code of Ethics. Our study does not involve human subjects, personally identifiable information, or any proprietary data. All datasets originate from publicly available resources that permit academic research use, and we release only derived tasks that preserve the original license conditions. The automated generation pipeline is designed to avoid creation of harmful or privacy-sensitive content and to prevent leakage of confidential information. All authors have read and agree to comply with the ICLR Code of Ethics.

## REPRODUCIBILITY STATEMENT

We make every effort to ensure full reproducibility of our results. Methods section details the multi-agent generation pipeline, verification mechanisms, and execution environment. Experiments section describes the evaluation protocol and model settings. Appendix lists all benchmark tasks and contains the exact agent prompts. An anonymized repository with source code and configuration files is provided in the supplementary materials to facilitate verification of all experiments.

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

## A  TASK, PROMPT AND STATISTICS DETAILS

### A.1  FULL LIST OF EVALUATION TASKS

Table 4 presents the raw dataset information of **Smith set** in dataset names, sizes and tags. The data sizes are relatively large to cover across different domains, modalities and formulations.

Table 4: Summary of Kaggle Competition Datasets

| Dataset Name | Size | Tags |
|---|---|---|
| **Vision–General** | | |
| veeralakrishna/200-bird-species-with-11788-images | 1.1 GB | universities and colleges, biology, online communities |
| sadhliroomyprime/cattle-weight-detection-model-dataset-12k | 44.1 GB | animals, business, agriculture, artificial intelligence, computer vision, pre-trained model |
| muhammetzahitaydn/hardhat-vest-dataset-v3 | 4.2 GB | intermediate, deep learning, public safety, yolo, object detection |
| balraj98/modelnet40-princeton-3d-object-dataset | 1.9 GB | earth and nature, science and technology |
| sunilthite/ovarian-cancer-classification-dataset | 3.3 GB | cancer, pre-trained model |
| iamtapendu/rsna-pneumonia-processed-dataset | 10.9 GB | healthcare, computer vision, image, image classification, image segmentation |
| pranavchandane/scut-fbp5500-v2-facial-beauty-scores | 1.1 GB | people, computer vision, cnn, image, regression |
| majdouline20/shapenetpart-dataset | 1.0 GB | computer science, classification, segmentation |
| thedatasith/sku110k-annotations | 13.2 GB | retail and shopping |
| tapakah68/supervisely-filtered-segmentation-person-dataset | 4.3 GB | arts and entertainment, people, computer vision, image |
| aletbm/urban-segmentation-isprs | 6.4 GB | earth and nature, data visualization, classification, image classification, image segmentation |
| hendrichscullen/vehide-dataset-automatic-vehicle-damage-detection | 2.1 GB | image, multiclass classification, insurance, object detection, segmentation |
| victorcallejasf/multimodal-hate-speech | 6.0 GB | nlp, image, multiclass classification, online communities, social networks |
| **Audio** | | |
| yashdogra/speech-commands | 2.3 GB | tensorflow, automatic speech recognition, speech synthesis, speech-to-text |
| daviddkarnowski/amateur-radio-transmissions-2-meter-fm-simplex | 34.0 GB | mobile and wireless, electronics, signal processing, audio, audio classification |
| soumendraprasad/sound-of-114-species-of-birds-till-2022 | 2.1 GB | arts and entertainment, earth and nature, beginner, intermediate, advanced, audio |
| mathurinache/the-lj-speech-dataset | 3.0 GB | artificial intelligence, advanced, signal processing, text, audio |
| chrisfilo/urbansound8k | 5.6 GB | arts and entertainment, music, classification, multiclass classification, audio |
| vjcalling/speaker-recognition-audio-dataset | 7.3 GB | arts and entertainment, music, classification, deep learning, audio |
| ikrbasak/sep-28k | 2.2 GB | healthcare, health, audio, numpy, scipy |
| abdelrahmanahmed110/quran-audio-dataset | 3.0 GB | music, religion and belief systems, audio |

| Competition Name | Size | Tags |
|---|---|---|
| raajanwankhade/oep-dataset | 11.0 GB | universities and colleges, computer vision, audio event classification, object detection, video classification |
| aryashah2k/noise-reduced-uaspeech-dysarthria-dataset | 8.0 GB | music, computer science, software, deep learning, audio synthesis, automatic speech recognition, audio classification, speech synthesis |
| jesusrequena/mlend-spoken-numerals | 1.1 GB | culture and humanities, languages, signal processing, audio |
| victorling/librispeech-clean | 28.1 GB | audio |
| imsparsh/deam-mediaeval-dataset-emotional-analysis-in-music | 1.8 GB | music, intermediate, advanced, multiclass classification, audio |
| vinayshanbhag/bird-song-dataset | 2.1 GB | music, audio |
| **NLP / Tabular** | | |
| devdope/900k-spotify | 1.0 GB | arts and entertainment, music, education, text generation |
| fayaznoor10/movie-transcripts-59k | 860.4 MB | arts and entertainment, movies and tv shows, nlp, text mining, multilabel classification |
| gowrishankarp/newspaper-text-summarization-cnn-dailymail | 503.3 MB | literature, nlp, text, news, transformers |
| nadyinky/sephora-products-and-skincare-reviews | 146.8 MB | computer science, nlp, recommender systems, retail and shopping, ratings and reviews |
| arshkon/linkedin-job-postings | 158.8 MB | employment, income, business, economics, nlp, jobs and career |
| sobhanmoosavi/us-traffic-congestions-2016-2022 | 2.3 GB | united states, categorical, transportation, tabular, urban planning |
| kgmuchiri/world-athletics-all-time-dataset | 52.9 MB | running, sports, data visualization, data analytics, tabular |
| edwardgaibor/pfaf-medical-plants-use-dataset | 13.9 MB | biology, agriculture, beginner, tabular, text |
| imoore/60k-stack-overflow-questions-with-quality-rate | 21.0 MB | music, nlp, text mining, text |
| spsayakpaul/arxiv-paper-abstracts | 44.6 MB | education, nlp, multilabel classification |
| arushchillar/disneyland-reviews | 11.1 MB | business, nlp, data visualization, tabular, ratings and reviews |
| simaanjali/emotion-analysis-based-on-text | 31.9 MB | earth and nature, nlp |
| jaidityachopra/esg-sustainability-reports-of-s-and-p-500-companies | 23.8 MB | nlp, investing, feature extraction, text preprocessing |
| smagnan/1-million-reddit-comments-from-40-subreddits | 71.2 MB | arts and entertainment, categorical, nlp, binary classification, online communities, social networks |
| salah1992/arabic-nli-pairs-multilingual-nli-26lang-2mil7 | 23.7 MB | earth and nature, linguistics, nlp, text, transformers, arabic |
| thedevastator/pubmed-article-summarization-dataset | 654.3 MB | bayesian statistics, earth and nature, nlp, text mining |
| shivamb/legal-citation-text-classification | 14.9 MB | australia, government, law, nlp, text |
| **Vision–Video** | | |

| Competition Name | Size | Tags |
|---|---|---|
| zaber666/meld-dataset | 11.0 GB | signal processing, text mining, text, audio, pre-trained model |
| rohanmallick/kinetics-train-5per | 33.3 GB | earth and nature, computer vision, deep learning, video, audio |
| matthewjansen/ucf101-action-recognition | 6.5 GB | computer vision, deep learning, video, transfer learning, video classification |
| rohitsuresh15/radroad-anomaly-detection | 7.3 GB | law, automobiles and vehicles, image, video, eyes and vision, urban planning |
| elin75/localized-audio-visual-deepfake-dataset-lav-df | 23.1 GB | advanced, video, audio |
| saberghaderi/-dfl-bundesliga-460-mp4-videos-in-30sec-csv | 10.1 GB | football, sports, science and technology, video, simulations |

## A.2 UNIFIED TASK STRUCTURE

The **Refactor** should deliver each task as a unified task format, specifically following the below directory structure. The assertions will ensure the existence of essential files and directories such as `prepare.py`, `metric.py`, `description.txt`, `sample_submission.csv`, `test_answer.csv`, `raw/`, `public/` and `private/`. Furthermore, assertions will ensure that the implementations of `prepare.py` and `metric.py` strictly follow the required format. Specifically, `prepare.py` must exactly implement a def prepare function whose input arguments include raw/, public/, and private/ directories. Likewise, `metric.py` must exactly implement a Metric class that inherits from the designated base class and provides the corresponding methods for task-aware submission validation and metric evaluation.

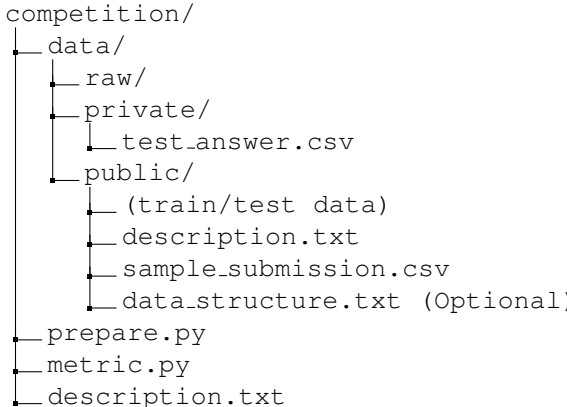

```
competition/
├── data/
│   ├── raw/
│   ├── private/
│   │   └── test_answer.csv
│   └── public/
│       ├── (train/test data)
│       ├── description.txt
│       ├── sample_submission.csv
│       └── data_structure.txt (Optional)
├── prepare.py
├── metric.py
└── description.txt
```

Figure 5: Unified directory structure that **Refactor** should deliver.

## A.3 NORMALIZATION DETAILS

For each task $t$ and model $m$, let $r_{t,m,u}$ denote the raw score from execution feedback at step $u \in \{1, \ldots, 10\}$. We define $\mathcal{D}_t \in \{+1, -1\}$ as the metric direction of task $t$, where $\mathcal{D}_t = +1$ indicates that higher metric values are better, and $\mathcal{D}_t = -1$ indicates that lower values are better.

The normalized score is computed as:

$$\tilde{r}_{t,m,u} = \begin{cases} \dfrac{r_{t,m,u} - \min_u r_{t,m,u}}{\max_u r_{t,m,u} - \min_u r_{t,m,u}}, & \mathcal{D}_t = +1, \\ \dfrac{\max_u r_{t,m,u} - r_{t,m,u}}{\max_u r_{t,m,u} - \min_u r_{t,m,u}}, & \mathcal{D}_t = -1. \end{cases}$$

If $\max r_{t,m} = \min r_{t,m}$, observed entries are set to 1 and missing ones to 0. We then forward-fill missing indices and compute a best-so-far trajectory via a prefix maximum:

$$y_{t,m,u} = \max\big(y_{t,m,u-1}, \tilde{r}_{t,m,u}\big).$$

This procedure yields a nondecreasing curve of length 10 per (task, model), which is then averaged pointwise across tasks to obtain category-level and overall trajectories.

## A.4 PROMPTS FOR MLE−SMITH AGENTS

We provide detailed prompts for MLE−Smith Agents in this section.

```
┌─────────────── Brainstormer Instruction ───────────────┐
You are an expert Kaggle competition designer. Your task is to
brainstorm diverse design choices for challenging, high-quality and
reasonable Kaggle competitions based on an existing dataset.

You are provided with detailed information about the dataset.
The dataset is already downloaded to the working directory with unzip.

You have access to tools for reading/writing files, listing directory
structure, and executing bash commands.

Always use function calls when you need to perform actions.
You can only call one function at a time.

Your work directory is {working_directory}, all actions and files
should be performed in this directory.

Use tools to explore the dataset to get insights.

Then based on insights from the dataset, brainstorm design choices for
challenging, high-quality and reasonable Kaggle competitions.

The design choice should include the following aspects,
be concise and informative:
- Concise problem overview: background, problem statement, and goal.
- Data utilization: for the given dataset,
  what data to use, what data to ignore.
- Data processing: how to process the data
- Metric: what metric to use, why it's fair and precise.
- Justification of the design choices: why the designed competitiion
  would be high-quality, challenging and solvable by ML techniques.
- Details of the ignored data: why the ignored data is not used,
  what information is missing.
- Difficulty level: how difficult the competition is,
  where the difficulty comes from.
- Tags: what tags the competition should be tagged with.

Principles:
- Only split the data into train and test sets.
- Ensure that only precise, reliable labels are used; no uncertain,
  ambiguous, or model-generated labels should be introduced.
- You must brainstorm and write at least one and
  at most {count} results. Determine the number of brainstorming outputs
  according to the intrinsic nature and properties of the dataset.
  - Some datasets are open-ended and naturally admit a wide range of
  tasks, while others are more specific and concentrated.
  - Aim to explore as many meaningful possibilities as the
  data genuinely supports, but do not force artificial variety|
  respect the dataset's natural boundaries.

Write your brainstorming results in "brainstorming\_i.md" file to
the working directory {working_directory},
```

```
is the index of the brainstorming result, from 1 to at most {count}.

## DATASET INFORMATION ##
{dataset_information}
```

---

```
┌──────────────────────┐
│ Designer Instruction │
└──────────────────────┘
You are an expert Kaggle competition designer. Your task is to create a
challenging, high-quality Kaggle competition based on existing dataset.

You are provided with detailed information about the dataset. And the
dataset is already downloaded to the working directory with unzip.

You have access to tools for reading/writing files, listing directory
structure, and executing bash commands.

Always use function calls when you need to perform actions.
You can only call one function at a time.

Your work directory is {working_directory},
all actions and files should be performed in this directory.

Now there is "brainstorming.md" file in the working directory,
pointing out the design direction for the competition.

## REQUIREMENTS ##
- Refer to "brainstorming.md" file for the design direction.
  But follow the requirements and instructions below.
- Make the competition challenging while maintaining a high quality
  standard.
- Participant should utilize ML techniques to solve the problem,
  including but not limited to:
    - Data Processing
    - Feature Engineering
    - Model Training
    - Model Evaluation
    ...
- Design the metric to be reasonable, fair and precise.
- Make good use of the data as possible, don't waste any good resources.
  Keep the scale rather than using subsets.
  No need to care about the runtime.
- Split the data into train/test sets appropriately.
- Always specify exactly the absolute path as arguments.
- All actions and files should be performed in the working directory.
- The competition should be challenging, but solvable by ML techniques.
- Make everything perfect rather than just trying to pass the tests.

A general pipeline for reference:

1. Utilize list_directory_structure tool to explore data structure.
2. Explore data files using the read_file tool;
   further extract files with bash commands if needed.
3. Design a concise and informative problem description
   and write it to "description.txt" in the working directory:
    - Include the problem statement,data description,evaluation metric,
      and any other relevant information.
    - Specify the final train/test data files for the competition,
      while don't specify the path of the data files.
    - Ignore timeline/prize/etc, they are not needed.
4. Write a "prepare.py" file:
    - Include complete train/test split process and
      sample_submission.csv generation
    - sample_submission.csv better has random but valid labels
```

        (same category as in test_answer.csv) rather than null values
- Test_answer and test_data (without the predicted labels) should be separated into two files, use "test_answer.csv" as the name
- Consider the correspondence between the test_answer and test_data
- Include detailed and comprehensive assert checks for the correctness of the split
- Specify the final train/test data files for the competition, align with the description.txt
- Validation set isn't needed, but keep it if it's already split
- The image, audio, and other related files should also be split together with the CSV files into train/test sets/folders.
- Rename files with names that might reveal their labels to avoid label leakage.
- Don't include data paths in csv files
- Set deterministic behavior for the split process.
- For classification tasks, all test labels should occur in training set at least once

5. Write a "metric.py" file, include functions to validate the format correctness of the submission and calculate the metric.
Deal with numerical values carefully to avoid nan/inf/etc.
6. Write a "test.py" script to test the correctness of the prepare.py and metric.py, run it to check the correctness until totally correct.
7. Optimize description.txt:
- No need to mention the original data files, only the final data files should be mentioned
- Take the view of a participant to review it (which means test_answer or irrelevant files shouldn't be mentioned) and make it perfect
- Make sure the competition is challenging, meaningful and solvable by ML techniques, and the metric is fair and precise.
- Make sure the description is informative, concise and accurate.
8. Optimize until all requirements are met with high quality (The test must pass).

## DATASET INFORMATION ##
{dataset_information}

---

$\boxed{\text{Refactor Instruction}}$

You are an expert Python developer. Your task is to refactor several Python files to meet some requirements.

You have access to tools for reading/writing files, listing directory structure, and executing bash commands.

You are provided with the working directory: {working_directory}, all actions and files should be performed in this directory.

All files you need are in the working directory. raw/ is where the data is downloaded and unzipped once.

samples/ directory is a good example, you can refer to it first to learn good practices and refactor the files to meet the requirements.

You may need to check the data files for details if needed.

## REQUIREMENTS ##
- You should finally refact metric.py and prepare.py to meet the requirements.
- metric.py should inherit from samples/base_metric.py and implement the abstract methods, give it a related name that ends with "Metrics", refer to samples/sample_metric.py for the implementation details.
  - "evaluate" and "validate_submission" should be implemented and

```
        aligned with "sample_submission.csv" and "test_answer.csv"
      - In addition to "self", "__init__" should have two arguments:
        "value" and "higher_is_better" (Determine the default);
        "evaluate" should have two arguments: "y_true" and "y_pred";
        "validate_submission" should have two arguments:
        "submission" and "ground_truth"
- prepare.py should implement exactly "def prepare(raw: Path,
  public: Path, private: Path)"
      - This function is a complete preparation process
      - Refer to samples/sample_prepare.py for the implementation details
      - Set deterministic behavior for the split process.
      - test_answer (participants shouldn't see) should be placed exactly
        in "private/" directory, other files (sample_submission, test/train
        data/images/audio/video/text/other, etc.) should be placed exactly
        in "public/" directory
- Write a comprehensive "test.py" script to test the correctness of
the prepare.py and metric.py, and run it to check the correctness.
Test "evaluate" and "validate_submission" of the metric.py with
"test_answer.csv" and "sample_submission.csv".
- Make sure the test.py is correct and comprehensive,
and the execution of test.py is correct.
- Don't include "main" function in metric.py and prepare.py
- Always specify exactly the absolute path as arguments.
- All actions and files should be performed in the working directory.
- Finally, there should be "private/", "public/", "samples/", "raw/"
directories, and "description.txt", "metric.py", "prepare.py",
"test.py" files in the working directory.
      - "raw/" directory should contain the original data files
      - "private/" directory should contain the test_answer.csv file
      - "public/" directory should contain the sample_submission.csv and
        all train/test data/images/audio/video/text/other files
        and description.txt. There should always be "test.csv"
        and "train.csv" in the "public/" directory if applicable.
      - Don't include or leak anything related to answers/golden labels
        in "public/" directory.
      - File directories in "description.txt" should be the same as the
        exact file directories in "public/" directory. Don't mention
        "private/" in the description.txt, only include files in "public/"
        directory.
      - "description.txt" is open to participants, so make it concise and
        informative, only include "public/" directory in the description.txt.
- Make everything perfect rather than just trying to pass the tests.
Optimize until all requirements are met with high quality
(The test must pass).
```

## A.5 DETAILS OF STATISTICAL MEASURES FOR ELO SET AGREEMENT

This section provides formal definitions, interpretation, and common use cases for all agreement statistics used to compare model-level Elo scores across different task sets.

### A.5.1 PEARSON LINEAR CORRELATION ($r$)

**Definition.** Given paired observations $\{(x_i, y_i)\}_{i=1}^{n}$,

$$r = \frac{\sum_{i=1}^{n}(x_i - \bar{x})(y_i - \bar{y})}{\sqrt{\sum_{i=1}^{n}(x_i - \bar{x})^2}\sqrt{\sum_{i=1}^{n}(y_i - \bar{y})^2}}.$$

**Meaning.** Measures the strength of *linear* association between two sets of scores. $r = 1$ indicates perfect positive linearity, $r = 0$ no linear association.

**Use.** Commonly used to assess whether two measurement methods produce proportionally similar values (e.g., Elo magnitudes across task sets).

### A.5.2 COEFFICIENT OF DETERMINATION ($R^2$)

**Definition.** For a simple linear regression $y_i = a + bx_i + \varepsilon_i$,

$$R^2 = 1 - \frac{\sum_i (y_i - \hat{y}_i)^2}{\sum_i (y_i - \bar{y})^2} = r^2 \quad \text{(for simple correlation)}.$$

**Meaning.** Represents the proportion of variance in $y$ explained by $x$. Higher $R^2$ indicates stronger predictive power of one set of scores for the other.

**Use.** Provides an intuitive measure of how much of the variability in Elo scores is shared between two task sets.

### A.5.3 SPEARMAN RANK CORRELATION ($\rho$)

**Definition.** Let $R(x_i)$ and $R(y_i)$ be the ranks of $x_i$ and $y_i$.

$$\rho = \frac{\sum_i (R(x_i) - \overline{R(x)})(R(y_i) - \overline{R(y)})}{\sqrt{\sum_i (R(x_i) - \overline{R(x)})^2}\sqrt{\sum_i (R(y_i) - \overline{R(y)})^2}}.$$

**Meaning.** Assesses whether the *ordering* of models is preserved, independent of absolute score scales.

**Use.** Robust to monotonic but nonlinear relationships, ideal for leaderboard stability checks.

### A.5.4 KENDALL RANK CORRELATION ($\tau_b$)

**Definition.** Let $C$ be the number of concordant pairs and $D$ the number of discordant pairs. Let $T_x$ and $T_y$ be the numbers of tied pairs in $x$ or $y$.

$$\tau_b = \frac{C - D}{\sqrt{(C + D + T_x)(C + D + T_y)}}.$$

**Meaning.** Quantifies pairwise ranking agreement while properly handling ties.

**Use.** Often preferred when ties occur (common in Elo ratings), providing a probabilistic interpretation: $\tau_b$ is the difference between the probability of concordance and discordance.

### A.5.5 TOP-$k$ OVERLAP

**Definition.** For a given $k$, let $S_x^k$ and $S_y^k$ be the sets of top-$k$ ranked items:

$$\text{Overlap}_k = \frac{|S_x^k \cap S_y^k|}{k}.$$

**Meaning.** Measures how consistently the *leaders* (top models) coincide.

**Use.** Highlights agreement in the most competitive region of leaderboards, which is often of primary interest.

### A.5.6 LIN'S CONCORDANCE CORRELATION COEFFICIENT (CCC)

**Definition.** Let $\mu_x, \mu_y$ be means, $\sigma_x^2, \sigma_y^2$ variances, and $\rho$ the Pearson correlation:

$$\text{CCC} = \frac{2\rho\sigma_x\sigma_y}{\sigma_x^2 + \sigma_y^2 + (\mu_x - \mu_y)^2}.$$

**Meaning.** Assesses both *precision* (correlation) and *accuracy* (closeness to the $45°$ identity line). A value of 1 indicates perfect agreement in both scale and location.

**Use.** Preferred when we need to verify numerical interchangeability beyond simple linear association.

### A.5.7 BLAND–ALTMAN ANALYSIS

**Definition.** For each pair $(x_i, y_i)$ compute

$$\text{Difference } d_i = x_i - y_i, \quad \text{Mean } m_i = \frac{x_i + y_i}{2}.$$

The plot of $d_i$ versus $m_i$ reveals systematic bias. The *limits of agreement* (LoA) are

$$\overline{d} \pm 1.96\, s_d,$$

where $\overline{d}$ is the mean difference and $s_d$ its standard deviation.

**Meaning.** Visualizes bias and scale discrepancies even when correlation is high.

**Use.** Widely used in clinical and experimental settings to test whether two measurement methods can be used interchangeably.

### A.5.8 CRONBACH'S $\alpha$

**Definition.** Suppose $k$ parallel measurements of the same quantity. Let $\sigma_t^2$ be the variance of the total score and $\sigma_j^2$ the variance of each measurement:

$$\alpha = \frac{k}{k-1}\left[1 - \frac{\sum_{j=1}^k \sigma_j^2}{\sigma_t^2}\right].$$

**Meaning.** Estimates internal consistency across multiple raters or measurement methods.

**Use.** Values above 0.9 indicate excellent reliability, supporting the claim that different Elo sets can be treated as interchangeable "raters" of model performance.

### A.5.9 INTRACLASS CORRELATION COEFFICIENT (ICC)

**Definition.** For the two-way random, absolute-agreement, single-measure model (denoted ICC$(2, 1)$):

$$\text{ICC}(2,1) = \frac{MS_B - MS_E}{MS_B + (k-1)MS_E + \frac{k}{n}(MS_R - MS_E)},$$

where $MS_B$ is the between-target mean square, $MS_R$ the between-rater mean square, $MS_E$ the residual mean square, $k$ the number of raters (here Elo sets), and $n$ the number of targets (models).

**Meaning.** Captures both correlation and absolute agreement among multiple raters.

**Use.** A high ICC confirms that Elo scores from different sets can be used interchangeably in downstream evaluations.

**Summary.** Together, these measures provide a comprehensive assessment of agreement, covering linear association, rank stability, numerical accuracy, and multi-rater reliability.

## B ABLATION STUDY

We conduct a comprehensive ablation study to justify the contribution of each component in the `MLE-Smith` pipeline.

### B.1 BRAINSTORMER

The Brainstormer increases the diversity of tasks generated from each dataset. To evaluate its contribution, we randomly select 10 datasets for which `MLE-Smith` had originally generated 3 tasks each. For each dataset, we regenerate 3 tasks in parallel *without* the Brainstormer and compare diversity in terms of prediction objectives and evaluation metrics (modality and domain are largely determined by the dataset itself).

Table 5 illustrates representative results. Without the Brainstormer, diversity degrades substantially: it is common to generate tasks with identical objectives and metrics (e.g., "dispensing category" repeated $3\times$ with the same F1 metric). In contrast, the Brainstormer produces tasks with distinct objectives and complementary metrics for the same dataset. An additional advantage is that the Brainstormer adaptively determines how many tasks to generate based on the dataset's intrinsic properties, rather than enforcing a fixed number.

Table 5: Task diversity with and without the Brainstormer (representative examples).

| Dataset | Objective | Metric |
|---|---|---|
| *With Brainstormer* | | |
| drugs-side-effects | User Rating (1–10) | macro-avg RMSE |
| | adverse event categories | macro-avg F1 |
| | safety category | Accuracy |
| mobile-uncleaned | market price | RMSLE |
| | Spec Score | MAE |
| | supports_5g | ROC-AUC |
| *Without Brainstormer* | | |
| drugs-side-effects | dispensing category ($\times$3) | macro-avg F1 |
| mobile-uncleaned | real-time price ($\times$3) | RMSLE |

### B.2 DESIGNER AND REFACTOR SEPARATION

The Designer generates task artifacts and the Refactor standardizes them into a unified format. We evaluate whether merging these into a single agent is preferable. The merged agent receives com-

bined prompts and must both design and standardize in one pass, with assertions applied only after completion. Both configurations are evaluated on 10 randomly sampled datasets.

Table 6: Comparison of separated vs. merged Designer–Refactor agents.

| Metric | Separated | Merged |
|---|---|---|
| Avg. retries | ∼1 (design) + ∼2 (refactor) | ∼9 total |
| Avg. steps | 11 + 19 = 30 | 26 (single context) |
| Avg. cost | $11.73 | $13.51 |

The separated design achieves a substantially lower failure rate (fewer retries), better cost efficiency, and avoids the compounding context-length problem of the merged approach. Separation also enables interleaved verification, allowing corrective actions at each stage rather than deferring all checks to the end.

### B.3 ASSERTIONS

Assertions serve as deterministic structural guards between pipeline stages. For the Designer stage, assertions flagged 4.4% of instances where required files (description, metric, prepare, test, sample_submission, etc.) were missing or malformed. For the Refactor stage, 9.6% of cases were flagged due to incomplete outputs, non-runnable artifacts, or violations of required interfaces (e.g., incorrect function signatures or failure to subclass the base metric class). Without assertions, these defects would propagate to downstream stages, wasting resources and causing cascading failures.

### B.4 REVIEWS

LLM-based Reviews complement assertions by detecting issues that cannot be captured by deterministic rules—for example, incomplete or misleading task descriptions, incorrect file identifiers in CSV references, filename-based label leakage, or conceptually unsound task designs. Reviews successfully detected 14.2% of issues that would be very difficult to identify through any other automated mechanism without substantial human expert effort.

### B.5 EXECUTION-BASED VALIDATION

Execution-based validation filters tasks by requiring an LLM agent to achieve a valid, non-trivial score. Tasks yielding perfect performance (e.g., accuracy = 1.0 or loss = 0.0) are discarded as insufficiently difficult or practically irrelevant. This stage filtered 11.2% of generated tasks, contributing significantly to the final quality of the benchmark.

### B.6 SUMMARY

Each component of the multi-agent design and verification pipeline serves an essential and complementary role. The Brainstormer ensures diversity, the Designer–Refactor separation ensures efficiency and modularity, Assertions prevent structural defects, Reviews catch semantic issues, and Execution-based Validation confirms empirical solvability. Together, they achieve an effective balance of cost, reliability, and task quality.

## C DIVERSITY OF PROBLEM-SOLVING STRATEGIES

We analyze the models and data processing techniques employed by agents when solving `MLE-Smith` tasks, providing evidence that the tasks require genuinely diverse problem-solving strategies.

### C.1 SOLUTION MODEL DISTRIBUTION

Table 7 summarizes the distribution of model categories used by agents across all tasks.

Table 7: Distribution of solution model categories on `MLE-Smith` tasks.

| Category | Percentage |
|---|---|
| Custom Models (NN / CNN / RNN / Seq2Seq / UNet / etc.) | ∼40% |
| Traditional ML (LogReg / Ridge / SVM / Boosting / etc.) | ∼25% |
| Vision – CNN Backbones (ResNet / EfficientNet / ConvNeXt) | ∼17% |
| NLP Transformers – Classification (BERT / RoBERTa / etc.) | ∼6% |
| Vision – Detection (Faster R-CNN / YOLO / RetinaNet) | ∼4% |
| Vision – Segmentation (UNet / DeepLab / FCN) | ∼3% |
| Audio / Speech (Wav2Vec2 / Whisper / HuBERT) | ∼2–3% |
| Vision – Video (R3D / R2Plus1D / Swin3D) | ∼2% |
| NLP Transformers – Generation (T5 / BART / Pegasus) | ∼2% |
| NLP Transformers – Other | ∼1% |

## C.2  DATA PROCESSING TECHNIQUES

Table 8 shows representative data processing operations and their frequency of use.

Table 8: Representative data processing operations used by agents.

| Category | Operation | Freq. (%) |
|---|---|---|
| General / Tabular | Data aggregation / grouping | 3.61 |
| | Feature engineering | 0.84 |
| | Class weight balancing | 0.51 |
| | Cross-validation | 0.37 |
| Text | Tokenization | 2.38 |
| | TF-IDF vectorization | 1.36 |
| | Text cleaning | 0.75 |
| Vision | Image normalization | 3.54 |
| | Image augmentation | 2.46 |
| Audio | Audio resampling | 1.34 |
| | Mel spectrogram generation | 0.73 |
| | Audio normalization | 0.61 |

The breadth of models and processing techniques confirms that `MLE-Smith` tasks demand flexible, task-specific strategies spanning multiple modalities and complexity levels.

## D  VERIFICATION PIPELINE DETAILS

We provide additional details on the hybrid verification mechanism to support reproducibility.

### D.1  ASSERTIONS: DETAILED CHECKS

**Pre-Refactor Assertions (after Designer).**  These assertions verify:

- Existence of all required files: `description.txt`, `metric.py`, `prepare.py`, `test.py`, `sample_submission.csv`, and `test_answer.csv`.
- Successful execution of `prepare.py` and `metric.py` without runtime errors.
- Successful generation of both a sample submission and a corresponding test answer.

**Post-Refactor Assertions.**  These assertions additionally enforce:

- Full conformance to the unified directory structure (Figure 5).

- Correct function signatures: `prepare.py` must implement exactly `def prepare(raw, public, private)`; `metric.py` must implement a `Metric` class inheriting from the designated base class with the required `evaluate` and `validate_submission` methods.
- Successful end-to-end execution of the test script.

## D.2 DISTINCTION BETWEEN REFACTOR AND ASSERTIONS

The **Refactor** module is an *active transformation stage* that rewrites and standardizes task artifacts produced by the Designer into a unified format. It restructures files, enforces consistent interfaces, and ensures cross-file coherence. In contrast, **Assertions** are *purely verification checks* that never modify files. They serve as deterministic gatekeepers, confirming that the produced artifacts are structurally valid and automation-ready. In short: Refactor *builds* the standardized structure; Assertions *verify* that the structure meets all rules.

## D.3 REVIEWS: COMMON ERROR TYPES

LLM-based Reviews have detected the following categories of issues:

- **Incomplete descriptions**: task descriptions omitting necessary information about data format, evaluation criteria, or submission requirements.
- **File reference errors**: incorrect file identifiers listed in CSV files or description text.
- **Label leakage**: filenames or directory structures that inadvertently reveal ground-truth labels.
- **Conceptual design flaws**: task setups that are technically executable but conceptually unsound (e.g., predicting a feature that is trivially derivable from other columns).

## D.4 EXECUTION-BASED VALIDATION: FILTERING CRITERIA

The validation agent is instructed to solve each task using meaningful ML methods and is prohibited from submitting trivial (sample) solutions. A task is retained only if:

- The full pipeline (data preparation, model training, evaluation, scoring) executes successfully without human assistance.
- The agent achieves a valid, non-trivial score.
- The score does not indicate a degenerate task (e.g., accuracy = 1.0 or loss = 0.0 are discarded as indicating insufficient difficulty or data leakage).

# E APPLICABILITY TO RAW AND UNSTRUCTURED DATASETS

To demonstrate that `MLE-Smith` generalizes beyond well-structured competition datasets, we evaluate it on three categories of notably raw data sources.

## E.1 UNPROCESSED TABULAR DATA: META KAGGLE

Meta Kaggle[1] consists of 41 unprocessed CSV files (44.28 GB in total), containing comprehensive metadata on Kaggle datasets, competitions, and notebooks. There are no predefined features or prediction targets, making this a challenging and genuinely raw tabular dataset. `MLE-Smith` generates 3 diverse, validated tasks (as determined by the Brainstormer):

## E.2 RAW SERVER LOGS: LOGHUB LINUX

The Loghub Linux dataset[2] contains raw syslog messages collected from `/var/log/messages` on a Linux server over a period of 260+ days. `MLE-Smith` generates 1 validated task:

---

[1] https://www.kaggle.com/datasets/kaggle/meta-kaggle/data
[2] https://github.com/logpai/loghub

Table 9: Tasks generated from the Meta Kaggle dataset.

| Objective | Metric | Description |
|---|---|---|
| Total downloads | RMSLE | Predict the eventual total number of downloads a dataset will receive given metadata at creation time. |
| TotalVotes | RMSLE | Predict each notebook's eventual community popularity measured by its TotalVotes. |
| Number of competitors | sMAPE | Predict the final number of competitors in a Kaggle competition using only information available at announcement time. |

Table 10: Task generated from the Loghub Linux dataset.

| Objective | Metric | Description |
|---|---|---|
| Event template IDs | Macro-avg F1 | Classify raw Linux syslog messages into fine-grained event template IDs given log metadata and free-text content. A challenging multi-class text classification problem with hundreds of distinct templates and strong class imbalance. |

### E.3    RAW SCIENTIFIC DATA: WEARABLE DEVICE DATASET

The Wearable Device Dataset[3] contains physiological recordings from induced stress and structured exercise sessions captured by a research-grade wearable device. `MLE-Smith` generates 1 validated task:

Table 11: Task generated from the Wearable Device dataset.

| Objective | Metric | Description |
|---|---|---|
| Physiological condition | Macro-avg F1 | Classify the physiological condition of a recording session into three classes based on wearable sensor data. |

### E.4    DISCUSSION

These results confirm that `MLE-Smith` can autonomously organize raw datasets and define appropriate features and labels, even when the data are complex and lack predefined structure. Representative examples of similarly raw datasets already present in the main benchmark include `900k-spotify`, `movie-transcripts59k`, `linkedin-job-postings`, and `us-traffic-congestions-2016-2022`, which similarly lack clearly defined prediction targets. We estimate that such raw datasets account for approximately 10–20% of the final benchmark.

---

[3]https://physionet.org/content/wearable-device-dataset/1.0.1/

