# OpenReview forum: "MLE-Smith: Scaling MLE Tasks with Automated Multi-agent Pipeline"
_ICLR.cc/2026/Conference — ICLR 2026 Poster_

### Official Review · Reviewer_Z7gA · 2025-10-25

**Soundness:** 2
**Presentation:** 2
**Contribution:** 2
**Rating:** 4
**Confidence:** 3

**Summary:**

This paper addresses the scarcity of high-quality, scalable training data for Machine Learning Engineering (MLE) tasks. The authors propose MLE-Smith, a fully automated multi-agent pipeline designed to convert raw datasets into competition-style MLE challenges. The pipeline operates on a "generate-verify-execute" paradigm, utilizing specialized agents to create tasks. A hybrid verification mechanism is employed to ensure structural integrity, semantic soundness, and empirical solvability. The authors apply this pipeline to 224 real-world datasets, generating 606 new MLE tasks.

**Strengths:**

1. The paper tackles an important problem. The bottleneck in creating high-quality, large-scale benchmarks for MLE agents is a real obstacle to progress in the field.

**Weaknesses:**

1. The contribution of the paper is limited. The proposed "generate-verify-execute" pipeline is a relatively common and established strategy for automated data synthesis and benchmark generation. Many prior works have employed similar paradigms, and the paper does not sufficiently articulate how this multi-agent application fundamentally differs from or improves upon those, beyond its application to the MLE domain.
2. While the verification mechanism ensures that tasks are executable and solvable (i.e., an agent can run code and get a non-trivial score), it does not appear to guarantee the correctness of their reference solutions. It is possible for a generated task to pass all verification checks while containing subtle logical flaws, or for the intended solution path to be suboptimal. The validation relies on "non-trivial predictive performance" rather than a strong guarantee of ground-truth correctness, which could impact the quality of the benchmark for training.
3. The experimental evaluation is lacking in-depth analysis. The results are largely aggregated, focusing on high-level Elo rankings and correlations. While this shows that the benchmark can rank models, it fails to provide insight into why models perform as they do or what specific challenges the new benchmark presents. The paper lacks a qualitative or fine-grained analysis of model failures. For example, what kinds of tasks are most difficult? What specific errors do top-performing agents make? Without this analysis, it is difficult to confirm the "challenging" nature of the benchmark, as it's unclear if it introduces new, harder problems or simply more of the same problems found in existing benchmarks.

**Questions:**

N/A

---

> ### Author Response · Authors · 2025-11-21
>
> We thank the reviewers for their valuable feedback and appreciation of our contributions. We address the main comments below.
>
> > **Weak 1: The Core Contributions**
>
> We would like to emphasize our core contributions.
>
> ### *1. A large, diverse, and verifiable MLE task dataset*
>
> We apply **MLE-Smith** to **224 real-world datasets** and automatically generate **606 competition-style MLE tasks** spanning a wide range of domains, objectives, and modalities. This represents one of the most comprehensive and scalable MLE task suites available.
>
> ---
>
> ### *2. A fully automated multi-agent pipeline that transforms raw data into high-quality MLE tasks*
>
> **MLE-Smith** operationalizes an efficient **generate–verify–execute** workflow that converts raw, heterogeneous datasets into standardized MLE challenges.
> Our pipeline focuses on solving this important and practical bottleneck in MLE training data: generating large quantities of **realistic**, **verifiable**, and **structurally consistent** tasks with **minimal/none human effort**.
>
> ---
>
> ### *3. A principled solution to the unique difficulty of verifying correctness in MLE task generation*
>
> Unlike domains such as **SWE** or **Web QA**, where tasks can be derived from existing repositories or established knowledge and where fully correct solutions generally exist, MLE tasks are constructed from raw datasets—including features, labels, metrics, and full problem formulations. Their correctness **cannot be assumed a priori**: both the problem definition and the solution space are **underdetermined**.
>
> **MLE-Smith** addresses this fundamental challenge through:
>
> - enforcing a **unified, verification-friendly output format**,
> - applying **multi-level structural and semantic verification**, and
> - validating **empirical solvability** through interactive execution.
>
> These mechanisms offer **generalizable design principles** for building reliable, scalable data-synthesis pipelines. We believe MLE-Smith can also provide useful insights for general task generation.
>
> > **Weak 2: The Correctness and Quality**
>
> Instead of relying on human experts to spend substantial time and effort on manual validation, MLE-Smith aims to maximize task quality through the multilayered automated process.
>
> - While “a strong guarantee of ground-truth correctness” is an ideal objective, in practice it is difficult to define rigorously and remains challenging even for human experts in real-world ML tasks. MLE-Smith substantially **reduces logical flaws and largely ensures** that generated tasks are both meaningful and solvable. We view the pursuit of even stronger automated correctness criteria as an important direction for future work, although we currently **do not see a clear automated method that would reliably improve task verification beyond this pipeline**.
>
> - Each task has its own data type, format, and distribution that reflect real-world applications. If an agent can achieve non-trivial predictive performance on such a task (neither perfect nor degenerate), and can further improve performance through better modeling, then the task would be useful for training agents.
>
> ---
>
> ### *Human Expert Evaluation:*
>
> We further conduct human evaluation to validate the correctness. Because the total number of generated tasks is large and manual inspection is extremely time- and labor-intensive, we sampled 50 tasks for human evaluation with ML experts from institutes and labs. The inspection procedure consisted of three steps:
> - Re-running the full script to ensure the task is runnable.
> - Checking for logical flaws, such as unsolvable setups (e.g., labels in the test set never appearing in the training set) or tasks that allow trivial high-score solutions (e.g., constructing a submission directly from leaked information).
> - Implementing a solution or reviewing the Agent’s solution to verify that it is neither trivial nor unreasonable, and that the resulting metric or score is appropriate.
>
> We found that the overall task quality was satisfactory, with no major errors identified.
>
> Another feasible approach is to present a human expert with one real task and one generated task, with the prize information and timeline removed from the real task, and ask them to identify which one is authentic. We believe that the generated tasks are **difficult to distinguish from real ones**. In the future, we will continue to validate the generated tasks from multiple perspectives to further demonstrate their quality.

---

> ### Author Response · Authors · 2025-11-21
>
> > **Weak 3**
>
> While notions such as "difficulty" and "challenge" are inherently hard to define strictly, we provide several concrete analyses to demonstrate that the tasks generated by **MLE-Smith** are indeed somehow hard for agents to solve in practice.
>
> ### *Task Error Rate*
> The detailed task error rate table (summary represents the modality average, each modality sampled task due to context limit):
>
> | Competition | Validate_code (%) | Execute_code (%) | Overall (%) |
> |------------|-------------------|------------------|-------------|
> | **Vision–General Summary** | **41.34%** | **38.56%** | **39.75%** |
> | 200-bird-s... | 34.78% | 21.54% | 27.03% |
> | hardhat-ve... | 23.26% | 46.55% | 36.63% |
> | **Audio Summary** | **46.22%** | **40.72%** | **43.40%** |
> | speech-com... | 40.00% | 48.21% | 44.14% |
> | amateur-ra... | 55.10% | 75.00% | 65.35% |
> | **NLP/Tabular Summary** | **33.43%** | **32.76%** | **33.06%** |
> | movie-tran... | 11.11% | 19.64% | 16.30% |
> | newspaper-... | 23.68% | 35.71% | 30.00% |
> | **Vision–Video Summary** | **49.79%** | **39.38%** | **44.38%** |
> | meld-data... | 30.00% | 32.31% | 31.43% |
> | kinetics-t... | 59.62% | 58.33% | 59.00% |
>
> Results indicate:
> - **High empirical error rates.**
>    Agents exhibit nontrivial and in many cases higher error rates on MLE-Smith tasks **compared with those in MLE-Dojo**. This arises for two reasons:
>    (1) the newly generated tasks are ones that the agents have likely **never encountered before** (even if they’ve seen the datasets before, the setting/description is totally different) , which forces them to rely on trial and error; and
>    (2) the tasks require **careful implementation** across all components of the pipeline including data processing, model design, and training, so it is easy for agents to fail at any step.
>
> - **Modalities that expose new weaknesses.**
>    Agents make more errors on video and audio tasks. These two modalities were underrepresented in prior benchmarks, and their inclusion meaningfully expands the problem space and introduces forms of complexity that existing benchmarks did not test.
>
>
> ---
> ### *Error Types*
>
> We also categorized different error types and presented the detailed error distribution below.
> Agents encounter a wide range of diverse errors when attempting to solve these tasks.
> | Category                   | Percentage |
> |----------------------------|------------|
> | **Value Error**            | 14.00%     |
> | **Data Processing Error**  | 13.62%     |
> | **Import Error**           | 12.72%     |
> | **File/Path Error**        | 11.03%     |
> | **Model/ML Error**         | 9.79%      |
> | **Submission Error**       | 7.78%      |
> | **Type Error**             | 6.63%      |
> | **Attribute Error**        | 6.31%      |
> | **Library API Error**      | 6.03%      |
> | **Syntax Error**           | 5.42%      |
> | **Unknown Category**       | 4.05%      |
> | **Key Error**              | 1.66%      |
> | **Memory/Resource Error**  | 0.89%      |
> | **Logic Error**            | 0.06%      |
>
>
> ---
>
> ### *Tasks require diverse problem-solving strategies*
>
> We conducted a detailed analysis of the models and data-processing methods used by the Agent when solving tasks.
>
> ### **Model Choices**
> | Category | Percentage |
> |--------------|----------------|
> | Custom Models (NN / CNN / RNN / Seq2Seq) | ~40% |
> | Traditional ML Models (LogReg / Ridge / SVM / KNN / Naive Bayes) | ~25% |
> | CNN Backbones (ResNet / EfficientNet / ConvNeXt / etc.) | ~17% |
> | Detection Models | ~4% |
> | Segmentation Models | ~3% |
> | Video Models | ~2% |
> | Text Classification (BERT / RoBERTa / etc.) | ~6% |
> | Text Generation (T5 / BART / etc.) | ~2% |
> | Audio / Speech Models | ~2–3% |
>
> ### **Data Processing Operations**
> | **General / Tabular** | % | **Text** | % | **Audio** | % |
> |-----------------------|---|----------|---|-----------|---|
> | Data aggregation / grouping | 3.61 | Tokenization | 2.38 | Audio resampling | 1.34 |
> | Feature engineering | 0.84 | TF-IDF | 1.36 | Mel spectrogram | 0.73 |
> | Feature selection | 0.30 | Text cleaning | 0.75  | Audio normalization | 0.61 |
> | Class weight balancing | 0.51 | N-gram generation | 0.45 | MFCC extraction | 0.45 |
> | Class weight computation | 0.40 | Text vectorization | 0.43 | — | — |
>
> The Agent employs a broad spectrum of models and processing techniques across modalities. The prevalence of **custom-built models** highlights the need for task-specific adaptation rather than reliance on fixed templates, demonstrating the challenging nature of tasks.
>
> Combining the above **analysis of error**, **Human Eval** and **diverse problem-solving strategies**, we believe that MLE-Smith indeed introduces novel and genuinely challenging problems that models cannot solve trivially. The quality of these tasks makes them well-suited for driving meaningful improvements in agent capabilities.

---

> ### Author Response · Authors · 2025-11-27
>
> Thank you again for your thoughtful engagement with our work. We truly appreciate the time and care you have put into your earlier comments.
>
> We hope our previous responses have adequately addressed the concerns you raised. When you have a moment, we would be grateful to know whether our clarifications resolved the issues you pointed out, or if there is anything further we could clarify or improve. Your feedback is extremely valuable to us, and we want to ensure that we have fully responded to your questions.
>
> Thank you very much for your time and consideration.

---

### Official Review · Reviewer_9fGm · 2025-10-27

**Soundness:** 3
**Presentation:** 3
**Contribution:** 4
**Rating:** 8
**Confidence:** 4

**Summary:**

Automating machine learning engineering (MLE) is an important task in LLM. However, the acquisition of high-quality MLE training data is difficult. In this work, the authors introduce MLE-smith, a fully automated multi-agent pipeline, to transform raw datasets into competition-style MLE challenges through an efficient generate–verify–execute paradigm for scaling MLE tasks with verifiable quality, real-world usability and rich diversity and a hybrid verification mechanism that enforces strict structural rules and high-level semantic soundness. The proposed benchmark utilizes 224 datasets to derive 606 tasks spanning multiple categories, objectives, and modalities.

**Strengths:**

1. The writing is clear and easy to understand.
2. The design of the pipeline is reasonable, detailed and economical. The pipeline incorporates a hybrid verification stack: deterministic assertions (format/structure-checks), semantic reviews (via agent), and execution-based validation (empirical solvability).  This provides multiple levels of guarantee that the generated tasks are structurally correct, semantically meaningful, and actually executable by agents while the cost is mere $2.11 per dataset. Given the cost, the scalability of this method is ensured.
3. They conduct comprehensive experiments on 8 different language models to testify to the quality of their dataset and their strong correlation with human-designed tasks.

**Weaknesses:**

1. I think the authors should include more details about the hybrid verification check in the Appendix, to ensure the reproducibility of this work.

**Questions:**

1. The difference of Refactor and Assertions seems ambiguous to me. It looks like that they are both executing formal checks. I would appreciate it if the authors could further clarify their differences, e.g. Refractor is for execution while Assertions is for check only?

---

> ### Author Response · Authors · 2025-11-21
>
> We thank the reviewers for their valuable feedback and appreciation of our contributions. We address the main comments below.
>
> > **Weak 1**
>
> We provide the details of the hybrid verification here, and we will surely include them in the appendix of the paper.
>
> - **Assertions**: Assertions ensure that each step has successfully produced all required output files, preventing downstream stages from being blocked by missing artifacts.
> For the Designer, the assertions verify the presence of required files—including description, metric, prepare, test, sample_submission and so on. These checks effectively identified cases where the Designer failed to produce a complete and correct file structure (4.4% of instances), enabling an immediate retry to generate the proper outputs before next steps. For the Refactor stage, 9.6% of cases were flagged due to incomplete outputs (missing files or directories), non-runnable artifacts (e.g., errors in prepare or test), or violations of required interfaces—such as incorrect function signatures in prepare or failure to properly subclass the base class in metric.
> In conclusion, Assertions prevents the pipeline from advancing with incomplete artifacts and avoids unnecessary time, resource waste, and downstream execution errors.
>
>
> - **Reviews**: While assertions enforce hard correctness constraints, certain issues cannot be reliably determined by fixed rules. For example, whether the description is complete and informative, whether file ids listed in csvs are correct, whether filenames leak label information, or whether the task design is conceptually sound. In these cases, LLM-based reviews are essential. We may update all common error types in the appendix to help readers gain more insights. We find that the reviews successfully detected 14.2% of issues that would be extremely difficult to identify through any other mechanism without substantial time and effort from human experts. Reviews and assertions are complementary, jointly providing an efficient, economical, and effective verification mechanism.
>
>
>
> - **Execution-based validation**: To verify that a task is truly reasonable and high-quality, the most direct method, aside from human experts spending substantial time and effort, is to let an LLM agent interact with it. In our setup, the agent is explicitly instructed (via prompt) to solve the task using meaningful methods such as training a model, and is prohibited from submitting trivial (sample) solutions. The agent must achieve a valid, non-trivial score, which confirms the task’s correctness (runnable with correct code and scripts). Furthermore, tasks are filtered by scores. For example, tasks that yield perfect performance (e.g., accuracy = 1.0 or loss = 0.0) are discarded, as they indicate insufficient task difficulty or practical relevance. Execution-based validation is able to filter 11.2% generated tasks, which contributes greatly to the quality of generated tasks.
>
>
> > **Question 1**
>
> The distinction between **Refactor** and **Assertions** is indeed important, and we are happy to clarify their roles in detail. We’ll make relevant contents clearer in the paper.
>
> ### *Refactor: Standardization and Structured Rewriting*
> The **Refactor** module is an *active transformation stage*. Its purpose is to **rewrite and standardize** the task produced by the Designer into a unified, standard format. This involves:
> - Restructuring files and directories
> - Enforcing consistent interfaces (e.g., `prepare.py`, `metric.py`)
> - Ensuring cross-file coherence
> - Applying the shared schema that all tasks must follow
>
> In short, it *modifies* the task artifacts to ensure every task conforms to the required task format.
>
> ---
>
> ### *Assertions: Deterministic Structural Validation*
> By contrast, **Assertions are purely verification checks**. They never modify files. Instead, they serve as rigid, deterministic guards that verify structural correctness:
>
> - Checking that files exist and follow the required layout
> - Ensuring `prepare.py` and `metric.py` run without errors
> - Validating function signatures and interfaces
> - Ensuring the directory fully matches the schema after Refactor
>
> Assertions act as **gatekeepers**, confirming that the produced task is structurally valid and automation-ready.
>
> In short, they *verify* correctness but never rewrite anything.

---

> ### Comment · Reviewer_9fGm · 2025-11-24
>
> Thank you for your response.
> After carefully reading both your reply and the questions raised by the other reviewers, I believe my initial concerns have been adequately addressed.
>
> At the same time, some of the points raised in the broader discussion appear somewhat unreasonable to me. To put it plainly, I do not think there exists any method that can comprehensively guarantee a completely flawless solution, given that this is a well-known NP-hard problem.
>
> I do, however, have an additional question for the authors. As reviewer 5zJ5 noted, the ability to process raw datasets is fundamental. Therefore, I am curious about the following:
>
> 1.	Have you incorporated the datasets mentioned in your rebuttal into your benchmark?
> 2.	What proportion do these datasets represent in the final benchmark?
>
> I will keep tuned to this rebuttal to check the concerns raised by other reviewers. Currently, I will keep my score.

---

> > ### Author Response · Authors · 2025-11-25
> >
> > Thank you very much for your thoughtful follow-up. We sincerely appreciate the time and care you’ve taken in reading both our responses and the broader discussion. We are grateful for your acknowledgement that the initial concerns have been adequately addressed. We also appreciate your perspective regarding the inherent difficulty of achieving a fully flawless solution.
> >
> > We provide the details about the following questions:
> >
> > > **Have you incorporated the datasets mentioned in your rebuttal into your benchmark?**
> >
> > Yes. The benchmark tasks include tasks generated from “raw” datasets, although they are not exactly the same ones mentioned in our rebuttal (the rebuttal examples were provided as additional experiments specifically addressing the reviewer’s requests regarding dataset types). As we stated in the rebuttal, *“We also note that Kaggle datasets are not universally well-structured or pre-cleaned.”* Consistent with this point, our benchmark does incorporate such raw datasets. Examples include:
> >
> > - `devdope/900k-spotify`
> > - `fayaznoor10/movie-transcripts59k`
> > - `arshkon/linkedin-job-postings`
> > - `sobhanmoosavi/us-trafficcongestions-2016-2022`
> > - `raajanwankhade/oep-dataset`
> > - …
> >
> > These datasets typically exhibit the following characteristics:
> >
> > 1. **They do not provide clearly defined features or prediction targets**, requiring the agent to design them autonomously.
> > 2. **They contain a large volume of files** (many directories) or **large CSVs with numerous rows/columns**.
> >
> > Results from task generation and benchmarking show that **MLE-Smith is capable of generating valid tasks from this class of raw datasets**.
> >
> > ---
> >
> > > **What proportion do these datasets represent in the final benchmark?**
> >
> > We did not intentionally control or filter the proportion of such tasks. After a careful review, we estimate that these tasks account for *approximately 10%–20%* of the benchmark.

---

### Official Review · Reviewer_pQFv · 2025-10-27

**Soundness:** 2
**Presentation:** 4
**Contribution:** 3
**Rating:** 4
**Confidence:** 3

**Summary:**

This paper addresses the critical bottleneck in scaling the creation of high-quality Machine Learning Engineering (MLE) benchmarks, which are essential for developing and evaluating sophisticated AI agents. The key of this work is MLE-Smith, a fully automated, multi-agent pipeline designed to transform raw datasets into competition-style MLE challenges at scale. The authors demonstrate the efficacy of their system by applying MLE-Smith to 224 real-world datasets, successfully generating 606 fully verified tasks spanning a wide range of modalities, objectives, and domains.

**Strengths:**

1.	The manuscript is well-motivated and well-written. The paper tackles a problem of significant importance. The ability to automatically generate diverse, high-quality MLE benchmarks at scale would be a major catalyst for research in autonomous MLE agents. Moreover, the proposed generate-verify-execute pipeline is well-presented and can be easily understood.

2.	The authors successfully demonstrate the system's capability at scale. They generate 606 tasks from 224 diverse datasets. The subsequent evaluation is extensive, involving eight different LLMs on a 100-task benchmark.

**Weaknesses:**

1.	The paper's main justification for task quality is that its performance rankings correlate with the MLE-Dojo benchmark. While this is a useful check, it feels like a narrow definition of "quality." Relying only on this metric makes it hard to judge other important aspects, like whether the tasks are truly novel, realistic, or test a wide range of skills. The paper's claims would be much more convincing if backed by more evidence, such as qualitative feedback from human MLE experts or an analysis showing that the tasks require diverse problem-solving strategies.

2.	The paper proposes a sophisticated system with multiple components, but there are no ablation studies to show what each part is actually contributing. For example, how much does performance change if you alter the agent's prompt or remove a specific reasoning step? Without this analysis, it's hard for the reader to know which components are essential to the system's success and which might be less important.

**Questions:**

1.	Could you consider supplementing the benchmark correlation with qualitative feedback from human MLE experts to provide a more holistic validation of task quality?

2.	Could you provide further analysis on the diversity of skills or problem-solving strategies required by your tasks, beyond the performance ranking correlation?

3.	How sensitive is the system's performance to changes in key components, such as the agent's prompt or the removal of a specific reasoning step?

---

> ### Author Response · Authors · 2025-11-21
>
> We thank the reviewers for their valuable feedback and appreciation of our contributions. We address the main comments below.
>
> > **Weak 1 & Question 1 & Question 2**
>
> Beyond the demonstrated correspondence to real Kaggle competitions, we present the following additional dimensions of analysis to further validate the quality of tasks generated by MLE-Smith.
>
> ---
>
> ### *Tasks require diverse problem-solving strategies*:
>
> - We conducted a detailed analysis of the models used by the Agent when solving tasks (summarized in the table below).
> | Category | Percentage |
> |--------------|----------------|
> | Custom Models (NN / CNN / RNN / Seq2Seq) | ~40% |
> | Traditional ML Models (LogReg / Ridge / SVM / KNN / Naive Bayes) | ~25% |
> | CNN Backbones (ResNet Series / EfficientNet Series / ConvNeXt / Others) | ~17% |
> | Detection (Faster R-CNN / RetinaNet / YOLO / related) | ~4% |
> | Segmentation (UNet / DeepLab / FCN) | ~3% |
> | Video Models (R3D / R2Plus1D / MC3 / X3D / Swin3D) | ~2% |
> | Classification (BERT / RoBERTa / DistilBERT) | ~6% |
> | Generation (T5 / BART / Pegasus / Seq2SeqLM) | ~2% |
> | NLP Other (MCQ, Emotion Models, Sentence Transformers, etc.) | ~1% |
> | Audio / Speech Models (Wav2Vec2 / Whisper / Hubert / WavLM)** | ~2–3% |
>
> - We also conducted a detailed analysis of the data processing methods used by the Agent when solving tasks (summarized in the table below).
> | Category / Operation               | Percentage | Category / Operation             | Percentage | Category / Operation                 | Percentage | Category / Operation              | Percentage |
> |-----------------------------------|------------|----------------------------------|------------|---------------------------------------|------------|----------------------------------|------------|
> | **General / Tabular**             |            | **Text**                         |            | **Vision**                             |            | **Audio**                         |            |
> | Data aggregation / grouping       | 3.61%      | Tokenization                     | 2.38%      | Image normalization                    | 3.54%      | Audio resampling                  | 1.34%      |
> | Feature engineering               | 0.84%      | TF-IDF vectorization             | 1.36%      | Image augmentation                     | 2.46%      | Mel spectrogram generation        | 0.73%      |
> | Feature selection                 | 0.30%      | Text cleaning                    | 0.75%      | Data augmentation                      | 0.65%      | Audio normalization               | 0.61%      |
> | Class weight balancing            | 0.51%      | N-gram generation                | 0.45%      |                                       |            | MFCC extraction                   | 0.45%      |
> | Class weight computation          | 0.40%      | Text vectorization               | 0.43%      |                                       |            |                                  |            |
>
>  The results show that the selected models/data processing methods span multiple modalities and cover a wide range of complexity, from simple regression models to more advanced architectures. They also require flexible adaptation to the specific task: a substantial fraction are Custom models rather than fixed approaches. This diversity in some aspects even **exceeds the MLE-Dojo benchmark**, particularly with respect to modality coverage (e.g., video and audio).
>
> ---
>
> ### *Human Expert Evaluation:*
>
> Because the total number of generated tasks is large and manual inspection is extremely time- and labor-intensive, we sampled 50 tasks for human evaluation with ML experts from institutes and labs. The inspection procedure consisted of three steps:
> - Re-running the full script to ensure the task is runnable.
> - Checking for logical flaws, such as unsolvable setups (e.g., labels in the test set never appearing in the training set) or tasks that allow trivial high-score solutions (e.g., constructing a submission directly from leaked information).
> - Implementing a solution or reviewing the Agent’s solution to verify that it is neither trivial nor unreasonable, and that the resulting metric or score is appropriate.
>
> We found that the overall task quality was satisfactory, with no major errors identified.
>
> Another feasible approach is to present a human expert with one real task and one generated task, with the prize information and timeline removed from the real task, and ask them to identify which one is authentic. We believe that the generated tasks are **difficult to distinguish from real ones**. In the future, we will continue to validate the generated tasks from multiple perspectives to further demonstrate their quality.

---

> ### Author Response · Authors · 2025-11-21
>
> > **Weak 2 & Question 3**
>
> We conducted a comprehensive ablation study to provide insights about each individual component. And we’ll surely include the ablation study in the paper.
>
> ---
>
>
> **Brainstormer**: The core role of Brainstormer is to increase the **diversity** of tasks that can be generated from a dataset, so that each dataset’s value can be fully utilized. We evaluated the pipeline without Brainstormer. Specifically, we randomly selected 10 datasets for which MLE-Smith had generated 3 tasks. For each selected dataset, we again generated 3 tasks in parallel without Brainstormer. For the analysis of diversity, since modality and domain are largely determined by the dataset itself, we focus primarily on the **objective** and the **metric**.
> The table below provides an intuitive view of part of the results due to context limit. As shown, **without the brainstormer, diversity is noticeably worse, and it is common to generate tasks with identical objective/metric**.
>
> - With Brainstormer:
> | Dataset                  | Objective                 | Metric                |
> |-------------------------|---------------------------|-----------------------|
> | drugs-side-effects… | User Rating (1–10)        | macro-averaged RMSE   |
> | *                       | adverse event categories  | macro-averaged F1     |
> | *                       | safety category           | Accuracy              |
> | mobile-uncleaned-data… | market price            | RMSLE                 |
> | *                       | Spec Score                | MAE                   |
> | *                       | supports_5g               | ROC-AUC               |
> | trending-movies…    | vote_count                | RMSLE                 |
> | *                       | liked                     | Macro F1              |
> | *                       | is_top_10pct              | Average Precision     |
>
> - Without Brainstormer:
> | Dataset                  | Objectives (Grouped)                           | Metric(s)              |
> |-------------------------|--------------------------------------------------|-------------------------|
> | drugs-side-effects… | dispensing category (repeated 3×)                | macro-averaged F1       |
> | mobile-uncleaned-data… | real-time price (repeated 3×)                | RMSLE                   |
> | trending-movies…   | popularity score (2×)                           | RMSLE                   |
> | *                       | popularity, vote                                 | averaged RMSLE          |
>
>
> Another advantage of this design is that the brainstormer allows the agent itself to decide how many tasks should be generated from a given dataset with insights it obtains after carefully exploring the dataset through tool use, rather than enforcing a fixed, non-reasonable number. This provides an economical and efficient solution to the diversity problem.
>
> ---
>
> **Designer and Refactor**:  The Designer and the Refactor are the core components of task generation. The functions performed by these two components are **essential**, so the ablation study focuses on whether they ought to be **merged into a single agent or kept as separate agents**.
> We made **three** modifications to merge the two components:
>
> - We combined their prompts so that the agent must both design and refactor.
>
> - Previously, assertions were applied after both; now, assertions are performed only after the entire process is completed.
>
> - Originally, the Designer and the Refactor each had three retry attempts. The combined module now has three retry attempts in total.
>
> We evaluate the merged agent on 10 randomly sampled datasets and compare it with the original two-agent setup.
> - *Retry Behavior*: **Unmerged:** ~1 retry in generation and ~2 in refactoring; **Merged:** ~9 retries in total, indicating a higher failure rate
>
> - *Step Usage*:  **Unmerged:** 11 steps for generation + 19 steps for refactoring; **Merged:** 26 steps in a single, longer context, making each step more expensive and failure-prone
>
> - *Cost*:
> **Unmerged:** \$11.73; **Merged:** \$13.51 (larger context + more retries)
>
> The results indicate that separating the two is more effective, efficient, and economical.
>
>
> ---
>
> **Assertions**: For the Designer, the assertions verify the presence of required files, including description, metric, prepare, test and so on. These checks effectively identified cases where the Designer failed to produce a complete and correct file structure (4.4% of instances). For the Refactor stage, 9.6% of cases were flagged due to incomplete outputs (missing files or directories), non-runnable artifacts (e.g., errors in prepare or test), or violations of required interfaces, such as incorrect function signatures in prepare or failure to properly subclass the base class in metric.

---

> ### Author Response · Authors · 2025-11-21
>
> **Reviews**: Certain issues cannot be reliably determined by fixed rules. For example, whether the description is complete and informative, or whether the task design is conceptually sound... In these cases, LLM-based reviews are essential. We find that the reviews successfully detected 14.2% of issues that would be very difficult to identify through any other mechanism without substantial time and effort from human experts.
>
> ---
>
> **Execution-based validation**: In our setup, the agent is explicitly instructed (via prompt) to solve the task using meaningful methods such as training a model, and is prohibited (via prompt) from submitting trivial (sample) solutions. The agent must achieve a valid, non-trivial score, which confirms the task’s correctness (runnable with correct code and scripts). Furthermore, tasks are filtered by scores. For example, tasks that yield perfect performance (e.g., accuracy = 1.0 or loss = 0.0) are discarded, as they indicate insufficient task difficulty or practical relevance. Execution-based validation is able to filter 11.2% generated tasks, which contributes greatly to the quality of generated tasks.
>
> ---
>
> **Conclusion**: Each component of the multi-agent design and the verification pipeline plays essential roles, aiming to achieve an effective balance of cost, performance, and efficiency. This also effectively addresses our core objective: alleviating the shortage of validated, high-quality MLE tasks.

---

> ### Comment · Reviewer_pQFv · 2025-11-26
>
> Dear Authors,
>
> Thank you for the detailed responses. Please incorporate the points discussed in the rebuttal into the revised version of the paper. As my concerns have now been addressed, I have decided to raise my score.

---

> > ### Author Response · Authors · 2025-11-26
> >
> > Thank you for your feedback. We sincerely appreciate your review, which is highly valuable and helpful to us. We will make sure to incorporate your raised points and the content from our rebuttal into the paper.

---

### Official Review · Reviewer_5zJ5 · 2025-11-01

**Soundness:** 2
**Presentation:** 3
**Contribution:** 2
**Rating:** 4
**Confidence:** 3

**Summary:**

The paper proposes MLE-Smith, a fully automated multi-agent system that converts raw datasets into competition-style machine learning engineering (MLE) tasks using a generate–verify–execute paradigm. Three specialized agents—Brainstormer, Designer, and Refactor—collaborate to create, standardize, and validate MLE tasks. Performance rankings on MLE-Smith tasks strongly correlate with those on human-curated benchmarks (MLE-Dojo), Pearson r ≈ 0.98, Spearman ρ ≈ 0.95, demonstrating high realism. Overall, MLE-Smith achieves scalable, diverse, and verifiable generation of MLE challenges for agent training and benchmarking.

**Strengths:**

1. Thorough evaluation – Quantitative correlation between synthetic and real tasks across multiple LLMs; diverse modalities and metrics.
2. Strong empirical realism – Demonstrated high rank and score correlation with human benchmarks, suggesting the tasks are faithful surrogates for real-world ones.
3. Reproducibility – Detailed description of environment, budgets, and execution setup; appendix lists datasets and code schema.

**Weaknesses:**

1. Ablation missing: The pipeline is explicitly broken into three agent roles (Brainstormer, Designer, Refactor) and a three-layer verification mechanism (Assertions, Reviews, Execution-based validation). However, it does not provide an empirical ablation study to justify the necessity of each individual component.
2. The pipeline was tested on 300 datasets from Kaggle. These datasets are typically well-structured and pre-cleaned for competition. It is unclear how MLE-Smith would handle "rawer" datasets (e.g., unstructured server logs, complex scientific data) that lack clear, pre-identified features or labels, and which may require significant domain expertise to formulate a task. In practice, those rawer data are even more useful since it requires good feature engineering strategies.
3. Lack of in-depth discussion: The paper states 807 tasks were generated and 606 were "fully verified". This implies a failure rate of ~25% (201 tasks). The paper does not provide a breakdown of why these tasks failed. clearer framing of scientific insight (why this works) would strengthen the whole paper as well.

**Questions:**

1. Could MLE-Smith handle domains outside Kaggle-like structured datasets?
2. Are there known failure patterns or bottlenecks in verification throughput? How many failed at the Assertions, Reviews, and Execution-based Validation stages, respectively?
3. What empirical benefit does the multi-agent separation provide versus a monolithic prompting approach?

---

> ### Author Response · Authors · 2025-11-21
>
> We thank the reviewers for their valuable feedback and appreciation of our contributions. We address the main comments below.
>
> > **Weak 1 & Question 2: Ablation study, Failure paterns/ratios in verification**
>
> We conducted a comprehensive ablation study to provide insights about each individual component. And we’ll surely include the ablation study in the paper.
>
> ---
>
>
> **Brainstormer**: The core role of Brainstormer is to increase the **diversity** of tasks that can be generated from a dataset, so that each dataset’s value can be fully utilized. We evaluated the pipeline without Brainstormer. Specifically, we randomly selected 10 datasets for which MLE-Smith had generated 3 tasks. For each selected dataset, we again generated 3 tasks in parallel without Brainstormer. For the analysis of diversity, since modality and domain are largely determined by the dataset itself, we focus primarily on the **objective** and the **metric**.
> The table below provides an intuitive view of part of the results due to context limit. As shown, **without the brainstormer, diversity is noticeably worse, and it is common to generate tasks with identical objective/metric**.
>
> - With Brainstormer:
> | Dataset                  | Objective                 | Metric                |
> |-------------------------|---------------------------|-----------------------|
> | drugs-side-effects… | User Rating (1–10)        | macro-averaged RMSE   |
> | *                       | adverse event categories  | macro-averaged F1     |
> | *                       | safety category           | Accuracy              |
> | mobile-uncleaned-data… | market price            | RMSLE                 |
> | *                       | Spec Score                | MAE                   |
> | *                       | supports_5g               | ROC-AUC               |
> | trending-movies…    | vote_count                | RMSLE                 |
> | *                       | liked                     | Macro F1              |
> | *                       | is_top_10pct              | Average Precision     |
>
> - Without Brainstormer:
> | Dataset                  | Objectives (Grouped)                           | Metric(s)              |
> |-------------------------|--------------------------------------------------|-------------------------|
> | drugs-side-effects… | dispensing category (repeated 3×)                | macro-averaged F1       |
> | mobile-uncleaned-data… | real-time price (repeated 3×)                | RMSLE                   |
> | trending-movies…   | popularity score (2×)                           | RMSLE                   |
> | *                       | popularity, vote                                 | averaged RMSLE          |
>
>
> Another advantage of this design is that the brainstormer allows the agent itself to decide how many tasks should be generated from a given dataset with insights it obtains after carefully exploring the dataset through tool use, rather than enforcing a fixed, non-reasonable number. This provides an economical and efficient solution to the diversity problem.
>
> ---
>
> **Designer and Refactor**:  The Designer and the Refactor are the core components of task generation. The functions performed by these two components are **essential**, so the ablation study focuses on whether they ought to be **merged into a single agent or kept as separate agents**.
> We made **three** modifications to merge the two components:
>
> - We combined their prompts so that the agent must both design and refactor.
>
> - Previously, assertions were applied after both; now, assertions are performed only after the entire process is completed.
>
> - Originally, the Designer and the Refactor each had three retry attempts. The combined module now has three retry attempts in total.
>
> We evaluate the merged agent on 10 randomly sampled datasets and compare it with the original two-agent setup.
> - *Retry Behavior*: **Unmerged:** ~1 retry in generation and ~2 in refactoring; **Merged:** ~9 retries in total, indicating a higher failure rate
>
> - *Step Usage*:  **Unmerged:** 11 steps for generation + 19 steps for refactoring; **Merged:** 26 steps in a single, longer context, making each step more expensive and failure-prone
>
> - *Cost*:
> **Unmerged:** \$11.73; **Merged:** \$13.51 (larger context + more retries)
>
> The results indicate that separating the two is more effective, efficient, and economical.
>
>
> ---
>
> **Assertions**: For the Designer, the assertions verify the presence of required files, including description, metric, prepare, test and so on. These checks effectively identified cases where the Designer failed to produce a complete and correct file structure (4.4% of instances). For the Refactor stage, 9.6% of cases were flagged due to incomplete outputs (missing files or directories), non-runnable artifacts (e.g., errors in prepare or test), or violations of required interfaces, such as incorrect function signatures in prepare or failure to properly subclass the base class in metric.

---

> ### Author Response · Authors · 2025-11-21
>
> **Reviews**: Certain issues cannot be reliably determined by fixed rules. For example, whether the description is complete and informative, or whether the task design is conceptually sound... In these cases, LLM-based reviews are essential. We find that the reviews successfully detected 14.2% of issues that would be very difficult to identify through any other mechanism without substantial time and effort from human experts.
>
> ---
>
> **Execution-based validation**: In our setup, the agent is explicitly instructed (via prompt) to solve the task using meaningful methods such as training a model, and is prohibited (via prompt) from submitting trivial (sample) solutions. The agent must achieve a valid, non-trivial score, which confirms the task’s correctness (runnable with correct code and scripts). Furthermore, tasks are filtered by scores. For example, tasks that yield perfect performance (e.g., accuracy = 1.0 or loss = 0.0) are discarded, as they indicate insufficient task difficulty or practical relevance. Execution-based validation is able to filter 11.2% generated tasks, which contributes greatly to the quality of generated tasks.
>
> ---
>
> **Conclusion**: Each component of the multi-agent design and the verification pipeline plays essential roles, aiming to achieve an effective balance of cost, performance, and efficiency. This also effectively addresses our core objective: alleviating the shortage of validated, high-quality MLE tasks.
>
> > **Weak 2 & Question 1: Handle Rawer Datasets**
>
> We identified several datasets that matches the description of “rawer” data and evaluated MLE-Smith on them: (1) unprocessed tabular data, (2) raw server logs, and (3) raw scientific data. We’ll surely update the results in the paper.
>
> ---
>
> - **Unprocessed tabular data: Meta Kaggle**
> (https://www.kaggle.com/datasets/kaggle/meta-kaggle/data) that consists of 41 unprocessed csv files (44.28 GB in total), including complete stats of kaggle. There are so many files and columns and no defined features/labels at all. So this is a very raw and challenging tabular dataset.
> MLE-Smith is able to generate 3 diverse, validated tasks (decided by Brainstormer):
> | Objective                     | Metric | TLDR |
> |-------------------------------|--------|------|
> | total number of downloads     | RMSLE  | “You are given metadata for Kaggle datasets at the time of their creation. Your task is to build a model that predicts the eventual total number of downloads a dataset will receive...” |
> | TotalVotes                    | RMSLE  | “Given information available at publish time for Kaggle notebooks ("Kernels"), predict each notebook’s eventual community popularity measured by its TotalVotes (upvotes)...” |
> | number of competitors         | sMAPE  | “Participants are challenged to predict the final number of competitors that will participate in a Kaggle competition using only information available when the competition is announced...” |
>
> ---
>
> - **Raw server logs: loghub Linux** (https://github.com/logpai/loghub). The dataset was collected from /var/log/messages on a Linux server over a period of 260+ days.
> MLE-Smith is able to generate 1 diverse, validated tasks (decided by Brainstormer):
> | Objective             | Metric              | TLDR |
> |-----------------------|---------------------|------|
> | event template IDs    | Macro-averaged F1   | “Participants must build a model that classifies raw Linux syslog messages into fine-grained event template IDs. Given the log metadata (Month, Date, Time, Level, Component, PID) ...” |
>
> ---
>
> - **Scientific raw data: Wearable Device Dataset from Induced Stress and Structured Exercise Sessions** (https://physionet.org/content/wearable-device-dataset/1.0.1/)
> MLE-Smith is able to generate 1 diverse, validated tasks (decided by Brainstormer):
> | Objective               | Metric            | TLDR |
> |-------------------------|-------------------|------|
> | physiological condition | Macro-averaged F1 | “Build a model that classifies the physiological condition of a recording session captured by a research-grade wearable device into three classes.” |
>
> We observed that MLE-Smith can autonomously organize raw datasets and define appropriate features and (or) labels, even when the data are relatively complex. We also note that Kaggle datasets are not universally well-structured or pre-cleaned. This demonstrates that MLE-Smith is capable of handling nontrivial, real-world raw data.
>
> > **Weak 3: Lack of in-depth discussion**
>
> The remaining 201 tasks were not included in the “fully verified” set at submission time because execution-based validation requires substantial GPU and CPU resources, and some generated tasks take a long time for an LLM agent to solve. As a result, we had not fully completed testing before submission. We will  update the final task count in the final version.

---

> ### Author Response · Authors · 2025-11-21
>
> > **Question 3: multi-agent separation provide versus a monolithic prompting approach**
>
> Compared with a monolithic prompting approach, MLE-Smith’s multi-agent design offers several advantages, which are also **validated by the empirical findings in ablation study above**:
>
> - **Higher success rate**. Decomposing the task into well-defined stages and validating each stage immediately greatly improves overall reliability, as discussed in Ablation Study above.
>
>
> - **Better efficiency**. Solving everything in a single prompt leads to extremely long contexts as the process progresses, which becomes inefficient and increases failure rates. More failures result in more retries, which further increases cost.
>
>
> - **Improved development and maintainability**. A single large prompt lacks transparency, making it difficult to identify where an error occurred or how to improve the system. In contrast, a multi-agent system provides clear modular boundaries, making development, debugging, and optimization much easier.
>
> - **Better integration with the verification pipeline**. A multi-agent system can interleave generation and verification, allowing corrective actions to be taken early rather than waiting until the entire task is produced before verifying, as in a single-prompt approach.

---

> ### Author Response · Authors · 2025-11-27
>
> Thank you again for your thoughtful engagement with our work. We truly appreciate the time and care you have put into your earlier comments.
>
> We hope our previous responses have adequately addressed the concerns you raised. When you have a moment, we would be grateful to know whether our clarifications resolved the issues you pointed out, or if there is anything further we could clarify or improve. Your feedback is extremely valuable to us, and we want to ensure that we have fully responded to your questions.
>
> Thank you very much for your time and consideration.

---

### Author Response · Authors · 2025-11-21
**Main Contributions**

We sincerely thank the reviewers for their valuable feedback and for recognizing the value of our contributions. We would also like to emphasize our core contributions.

### **1. A large, diverse, and verifiable MLE task dataset**

We apply **MLE-Smith** to **224 real-world datasets** and automatically generate **606 competition-style MLE tasks** spanning a wide range of domains, objectives, and modalities. This represents one of the most comprehensive and scalable MLE task suites available.

---

### **2. A fully automated multi-agent pipeline that transforms raw data into high-quality MLE tasks**

**MLE-Smith** operationalizes an efficient **generate–verify–execute** workflow that converts raw, heterogeneous datasets into standardized MLE challenges.
Our pipeline focuses on solving this important and practical bottleneck in MLE training data: generating large quantities of **realistic**, **verifiable**, and **structurally consistent** tasks with **minimal/none human effort**.

---

### **3. A principled solution to the unique difficulty of verifying correctness in MLE task generation**

Unlike domains such as **SWE** or **Web QA**, where tasks can be derived from existing repositories or established knowledge and where fully correct solutions generally exist, MLE tasks are constructed from raw datasets—including features, labels, metrics, and full problem formulations. Their correctness **cannot be assumed a priori**: both the problem definition and the solution space are **underdetermined**.

**MLE-Smith** addresses this fundamental challenge through:

- enforcing a **unified, verification-friendly output format**,
- applying **multi-level structural and semantic verification**, and
- validating **empirical solvability** through interactive execution.

These mechanisms offer **generalizable design principles** for building reliable, scalable data-synthesis pipelines. We believe MLE-Smith can also provide useful insights for general task generation.

---

### Meta-Review · Area_Chair_BbkW · 2026-01-04

**Summary:**

The reviewers broadly agree that MLE-Smith tackles an important and timely problem, i.e., scaling the creation of high-quality machine learning engineering (MLE) benchmarks. The paper proposes an automated, multi-agent (generate–verify–execute) pipeline and demonstrates scalability by generating hundreds of competition-style tasks from real-world datasets. Multiple reviewers acknowledge the system's solid engineering, clear presentation, and strong empirical evidence showing high correlation with human-curated benchmarks.

The main initial concerns centered on (i) lack of ablation studies to justify the multi-agent design and verification stack; (ii) limited discussion of failure modes and verification breakdowns; (iii) unclear ability to handle raw or non-Kaggle-style datasets; and (iv) whether benchmark quality is sufficiently validated beyond ranking correlation metric. Through a detailed rebuttal, the authors addressed most of these points by providing ablation analyses, verification statistics, experiments on raw datasets, human expert evaluation, and deeper error analyses.

However, one remained issue has not been well resolved. Regarding the comments from Reviewers Z7gA and 5zJ5, there is lack of a principled guarantee of task correctness beyond empirical solvability. The author's response appropriately acknowledges that strong ground-truth correctness is inherently hard for MLE tasks, but the rebuttal mainly reframes this as an unavoidable limitation rather than offering a stronger methodological mitigation. The offered human evaluation in author's rebuttal and multilayered verification reduce risk, but they do not change the underlying fact that correctness is defined operationally ("an agent can run code and get a non-trivial score") rather than formally or technically. Moreover, the human evaluation presented in the rebuttal is limited in scope, lacking statistical analysis and discussion of inter-annotator agreement or alignment.

Most actionable, technical concerns (ablations, verification clarity, raw data handling, quality validation) were convincingly addressed. The remaining issues are largely conceptual or aspirational (novelty framing, formal correctness guarantees, depth of analysis). Reviewer 9fGm maintained a positive assessment, and Reviewer pQFv decided to raise their score. Thus, I am recommending an acceptance but I wouldn't mind if the paper gets rejected.

**Reviewer Concerns:**

Concerns largely addressed:

* Lack of ablation studies for multi-agent design and verification stack (Reviewer 5zJ5, Reviewer pQFv)
* Unclear failure patterns and verification bottlenecks (Reviewer 5zJ5)
* Ability to handle raw, non-Kaggle-style datasets (Reviewer 5zJ5, Reviewer 9fGm)
* Ambiguity between Refactor and Assertions (Reviewer 9fGm)
* Narrow definition of task quality that only discusses rank correlation only (Reviewer pQFv)

Concerns partially or not fully addressed:

* Limited conceptual novelty of generate–verify–execute (Reviewer Z7gA). The authors reframed novelty as systematization and application to MLE, rather than proposing a new paradigm.
* Lack of strong guarantees of ground-truth correctness (Reviewer Z7gA). The authors argued that formal correctness guarantees are inherently difficult for MLE tasks and provided human evaluation as mitigation. The limitation is acknowledged rather than solved.
* Limited fine-grained analysis of model behavior and failure causes (Reviewer Z7gA). The author's rebuttal adds error-type and modality-level analyses. However, deeper qualitative insights (e.g., systematic failure modes or skill-specific weaknesses) remain limited.

**Reviewer Scores:**

Reviewer 5zJ5: change to 6 since almost all their concerns have been addressed.

Reviewer pQFv: change to 6 since the author has resolved their concerns and the reviewer has responded during rebuttal.

Reviewer 9fGm: remains the score as 8.

Reviewer Z7gA: remains the score as 4 since the author has not fully addressed their concerns.

---

### Decision · Program_Chairs · 2026-01-26

Accept (Poster)